# Inter-hemispheric seasonal comparison of Polar Amplification using radiative forcing of quadrupling CO$_2$ experiment

Fernanda Casagrande[1], Ronald Buss de Souza[1], Paulo Nobre[1], Andre Lanfer Marquez[1]

[1]Center for Weather Forecasting and Climate Studies, National Institute for Space Research, Cachoeira Paulista, 13620-000 Brazil.

*Correspondence to*: Fernanda Casagrande (Fernanda.casagrande@inpe.br)

**Abstract.** The numerical climate simulations from the Brazilian Earth System Model (BESM) are used here to investigate the response of Polar Regions to a forced increase of CO$_2$ (Abrupt-4xCO$_2$) and

compared with Coupled Model Intercomparison Project phases 5 (CMIP5) and 6 (CMIP6) simulations. The main objective here is to investigate the seasonality of the surface and vertical warming as well as the coupled processes underlying the polar amplification, as changes in sea ice cover. Polar Regions are described as the most climatically sensitive areas of the globe, with an enhanced warming occurring during the cold seasons. The asymmetry between the two poles is related to the thermal inertia and the

coupled ocean atmosphere processes involved. While in the northern high latitudes the amplified warming signal is associated to a positive snow and sea ice albedo feedback, for southern high latitudes the warming is related to a combination of ozone depletion and changes in the winds pattern. The numerical experiments conducted here demonstrated a very clear evidence of seasonality in the polar amplification response, as well as, linkage with sea ice changes. In winter, for the northern high

latitudes (southern high latitudes) the range of simulated polar warming varied from 10 K to 39 K (-0.5 K to 13 K). In summer, for northern high latitudes (southern high latitudes) the simulated warming varies from 0 K to 23 K (0.5 K to 14 K). The vertical profiles of air temperature indicated stronger warming at the surface, particularly for the Arctic region, suggesting that the albedo-sea ice feedback overlaps with the warming caused by meridional transport of heat in the atmosphere. The latitude of the

maximum warming was inversely correlated with changes in the sea ice within the model's control run. Three climate models were identified as having high polar amplification for the Arctic cold season (DJF): IPSL-CM6A-LR (CMIP6), HadGEM2-ES (CMIP5) and CanESM5 (CMIP6). For the Antarctic, in the cold season  (JJA), the climate model identified as having high polar amplification were: IPSL-CM6A-LR (CMIP6), CanESM5(CMIP6) and FGOALS-s2 (CMIP5). The large decrease in sea ice

concentration is more evident in models with great Polar Amplification, and for the same range of latitude ($75^{o}$ N – $90^{o}$ N). Also, we found, for models with enhanced warming, expressive changes in the sea ice annual amplitude with outstanding ice-free conditions from May to December (EC-Earth3-Veg) and June to December (HadGEM2-ES). We suggest that the large bias found among models can be related to the differences in each model to represent the feedback process and also as a consequence of each distinct sea ice initial conditions. The polar amplification phenomenon has been observed previously and is expected to become stronger in the coming decades. The consequences for the atmospheric and ocean circulation are still subject to intense debate in the scientific community.

## 1 Introduction

Polar regions have been shown to be more sensitive to climate change than the rest of the world (Smith et al., 2019; Serreze and Barry, 2011). The Arctic is warming at least twice as fast as the northern hemisphere and as the globe as a whole. This phenomenon is known as the Arctic Amplification (AA) and is combined with a fast shrinking of the sea ice cover (Serreze and Barry, 2011; Kumar et al., 2010; Screen and Simmonds, 2010). Previous research has indicated that the enhanced Arctic warming is a response to anthropogenic Greenhouse Gas (GHG) forcing, which, in turn, intensifies many complex non-linear coupled ocean-atmosphere feedbacks (e.g. the sea ice albedo feedback) (Stuecker et al., 2018; Pithan and Mauritsen, 2014; Alexeev et al., 2005). The sea ice-albedo feedback is one of the key mechanisms to amplify the Arctic warming, playing an important role in global climate change (Stuecker et al., 2018; Pithan and Mauritsen, 2014). In contrast to the Arctic sea ice, the total sea ice cover surrounding the Antarctic continent has increased in association with cooling over eastern Antarctica and warming over the Antarctic Peninsula. The physical ocean atmosphere coupled processes responsible for Antarctic sea ice rising are still unclear. Turner et al., (2017) show the unprecedented springtime retreat of Antarctic sea ice in 2016. However, results derived from numerical simulations and observations point to a combination of changes in the wind pattern, the ocean circulation, accelerated basal melting Antarctica's ice shelf and the ozone depletion (Marshall et al., 2014 Thompson et al., 2011; Bintanja et al., 2013; Thompson and Solomon, 2002). According to Marshall et al., (2014), these two-poles inter-hemispheric asymmetries strongly influence the Sea

Surface Temperature (SST) response to an increase in the global $CO_2$ forcing, accelerating the warming in the Arctic while delaying it in Antarctica.

Numerous scientific publications based on both, observations and state-of-the-art Global Climate Model simulations for the high latitudes of the northern hemisphere have shown that AA is an intrinsic feature of the Earth's climate system (Smith et al., 2019; Vaughan et al., 2013; Serreze and Barry, 2011; Screen and Simmonds, 2010). These works suggested that the Surface Air Temperature (SAT) will continue to increase with effects extending beyond the Arctic region (Dethloff et al., 2019; Smith et al., 2019; Holland and Bitz, 2003; Serreze and Barry, 2011; Winton 2006; Bintanja et al., 2013). Although the annual average SAT at northern mid- and high latitudes is increasing, the wintertime SAT has decreased since the 1990 (Zhang et al., 2016; Mori et al., 2014; Cohen et al., 2012; Honda et al., 2009).

Bekryaev et al., (2010), for instance, found a warming rate of $1.36^\circ C$ century$^{-1}$ for the period from 1875 to 2008 using an extensive set of observational data from meteorological stations located at high latitudes of the northern hemisphere ($> 60^\circ N$). That trend is almost double that of the northern hemisphere trend as a whole ($0.79^\circ C$ century$^{-1}$), with an accelerated warming rate in the most recent decade. Rigor et al., (2000) also using an observational dataset showed that the Arctic warming varies largely among regions and that changes in SAT are also related to the Arctic Oscillation (Ambaum et al., 2001).

The Arctic Ocean temperature and ocean heat fluxes also have increased over the past several decades (Walsh, 2014; Polyakov et al., 2010; Polyakov et al., 2008). According to Polyakov et al., (2017), the recent sea ice shrinking, weakening of the halocline and shoaling of the intermediate-deep Atlantic water masses layer in eastern Eurasia Basin have increased the winter ventilation in the ocean interior, making the region structurally similar to the western Eurasian Basin. The authors described these processes as an "Atlantification" phenomenon and represent an essential step toward a new Arctic climate state.

Holland and Bitz, (2003) using a set of 15 state-of-the-art CMIP models found that the range of simulated Arctic warming as response to a doubling of $CO_2$ concentration varies largely between the models ranging from 1.5 to 4.5 times the global mean warming. The large differences among the

models are related to differences in simulating the ocean's meridional heat transport, the polar cloud cover and the sea ice (e.g. a simulation with thinner sea ice cover presents a higher polar amplification).

According to Shu et al., (2015), Global Climate Models in general offer much better simulations for the Arctic than for the Antarctica. Turner et al., (2015) suggested that the main problem of climate models in the high latitudes of the southern hemisphere is their inability to reproduce the observed (although slight) increase in Sea Ice Extent (SIE). Bintanja et al., (2015) and Swart and Fyfe, (2013) have demonstrated the importance to include the effect of the increasing freshwater input from Antarctic continental ice into the Southern Ocean. The authors described that the ice sheet dynamics, essential for having accurate sea ice simulations, is currently disregarded in all CMIP5 models. Swart and Fyfe (2013) also suggested that this deficiency may significantly influence the simulated sea ice trend because the subsurface ocean warming causes basal ice-shelf melt, freshening the surface waters, which eventually leads to an increase in sea ice formation. Moreover, the instrumental network for data collection in Antarctica and the Southern Ocean is considered scarce (even more than in the Arctic), inhomogeneous and insufficiently dense to validate climate models. Therefore, for the high latitudes regions of the southern hemisphere, the effects of the ongoing climate change and its associated processes are still considered hot topics that lack conclusive answers.

How the polar climate will change as response to an external forcing deeply depends on feedback processes, which operates to amplify or diminish the effects of climate change forcing. These feedbacks depend on the integrated coupled processes between ocean-atmosphere-cryosphere over a large spectrum of spatial and temporal scales, which makes the quantification of them even more complicated.

Here the seasonal sensitivity of high latitudes as a response to quadrupling atmospheric $CO_2$ is investigated using the recently developed Brazilian Earth System Model, coupled ocean-atmosphere version 2.5 (BESM-OA V2.5) and comparing its results with those from 32 other Coupled General Circulation Models participating in CMIP5 and CMIP6. Our goal is to investigate the coupled processes underlying the polar warming by seasons. The paper is organized as follows: Section 2 provides a description of the climate models and experimental design[s] used in this work, focusing on the BESM-OA V2.5 model (Veiga et al., 2019; Giarolla et al., 2015; Nobre et al., 2013). In Section 3, the

seasonality in the surface warming in high latitudes is examined of both northern and southern hemispheres and results from different models are compared. Section 4 provides an analysis of the vertical structure of air temperature warming, spatial pattern of sea ice changes and a discussion about the coupled ocean atmosphere processes and feedback mechanisms involved. A summary of results and conclusions are presented in Section 5.

## 2 Data Sources

### 2.1 Numerical Design

This study used two numerical experiments from CMIP5 and CMIP6: (i) piControl: it runs for 700 years, forced by invariant pre-industrial atmospheric $CO_2$ concentration level (280ppmv) and (ii) Abrupt 4x$CO_2$: it runs for 460 years, comprising an abrupt instantaneous quadrupling of atmospheric $CO_2$ level concentration from the piControl simulation. The design of both experiments follows the CMIP5 protocol (Taylor et al., 2012) and Eyring et al. (2016) for CMIP6 numerical experiments.

Although an instantaneous quadrupling $CO_2$ scenario is not realistic for the 21st century compared with RCP scenarios and observations, this scenario can give us a measure of climate sensitivity and how large can be the response of the polar region in comparison to the globe as a whole. The results are compared for polar amplification (changes in air temperature) and sea ice cover, for the same numerical experiment.

For CMIP5 numerical experiment, the follow models are used: BESM-OA V2.5 (Nobre et al., 2013; Veiga et al., 2019), ACCESS-3 (Bie et al., 2013; Collier and Uhe, 2012), GFDL-ESM2M (Griffies, 2012), IPSL-CM5-LR (Dufresne et al., 2013), MIROC-ESM (Watanabe et al., 2011), MPI-ESM-LR (Stevens et al., 2013), NCAR-CCSM4 (Gent et al., 2011), CanESM2 (Chylek et al., 2011), FGOALS-s2 (Bao et al., 2013), GFDL-ESM2G (Delworth et al., 2006), GISS-E2_H (Schmidt et al., 2006), HadGEM2-ES (Collins et al., 2008), MIROC5(Watanabe et al., 2010), MPI-ESM-P(Giorgetta et al., 2013), MRI-CGCM3(Yukimoto et al., 2012).

For CMIP6 numerical experiments, the follow models are used: ACCESS-CM2 (Martin et al., 2019), CAMS-CSM1-0 (Rong, 2019), CanESM5 (Swart et al., 2019), CMCC-CM2-SR5 (Fogli et al.,

2020), CNRM-ESM2-1 (Roland., 2018), ACCESS-ESM1-5 (Ziehn et al., 2019), E3SM-1-0 (Bader et al., 2019), EC-Earth3-Veg, FGOALS-G3 (Li et al, 2019), GISS-E2-1-H (Schmidt et al., 2006), INM-CM4-8 (Volodin et al 2019), MIROC6 (Tatebe et al., 2018), MIROC-ES2L (Ohgaito et al., 2019), MPI-ESM1-2-LR (Fiedler et al., 2019), MRI-ESM2-0 (Yukimoto et al., 2019).

**1.2 Brazilian Earth System Model**

The Brazilian Earth System Model, Version 2.5 (BESM-OA2.5) used here is a global climate coupled ocean-atmosphere-sea ice model, and is part of CMIP5 project. The atmospheric component of BESM-OA2.5 is BAM (Brazilian Atmospheric Model) and was described in detail by Figueroa et al., (2016). The lastest version of BAM, used here and described by Figueroa et al., (2016) and Veiga et al., (2019), has spectral horizontal representation truncated at triangular wave number 62, grid resolution of approximately 1.875°×1.875°, and 28 sigma levels in the vertical, with unequal increments between the vertical levels (i.e., a T62L28). Two important changes were implemented on the BESM last version: (i) a new microphysics scheme, described by Ferrier et al., (2002) and Capistrano et al., (2020) and (ii) a new surface layer scheme, described by Capistrano et al., (2020) and Jimenez and Dudhia, (2012). These key changes represent an improvement in surface layer, resulting in better representation of near-surface air temperature, wind and humidity at 10 m. The main improvements occur over the ocean, where temperature, wind and humidity are important to calculate the heat fluxes at ocean-atmosphere-sea ice interface.

The oceanic component of BESM-OA2.5 is the Modular Ocean Model, Version 4p1, from National Oceanic and Atmospheric Administration-Geophysical Fluid Dynamics Laboratory (MOM4p1/NOAA-GFDL), described in detail by Griffies, (2009). The MOM4p1 includes a Sea Ice Simulator (SIS) built-in ice model (Winton 2000). The SIS has five ice thickness categories and three vertical layers (one snow and two ice). To calculate ice internal stresses are used the elastic-viscous-plastic technique described by Hunke and Dukowicz, (1997). The thermodynamics is given by a modified Semtner's three-layer scheme (Semtner, 1976). SIS is able to calculate sea ice concentrations, snow cover, thickness, brine content and temperature. The horizontal grid resolution of MOM4p1 in the longitudinal direction is a set to 1°. The latitudinal direction varies uniformly, in both hemispheres,

from 1⁄4$^{o}$ between 10$^{o}$ S and 10$^{o}$ N to 1$^{o}$ of resolution at 45$^{o}$ and to 2$^{o}$ of resolution at 90$^{o}$. The vertical axis has 50 levels (upper 220m, has 10 m resolution, increasing to about 360 at deeper levels. The MOM4p1 and BAM models were coupled using FMS coupler. FMS coupled was developed by NOAA-GFDL. The BAM model receives SST and ocean albedo from MOM4p1 and SIS (hour by hour). The MOM4p1 receives momentum fluxes, specific humidity, pressure, heat fluxes, vertical diffusion of velocity components and freshwater. The Monin-Obukhov scheme is used to calculate the wind stress fields (Obukov, 1971).

## 3 Results and Discussion

First we discuss the seasonality of polar warming near the surface in the Arctic, vertical profile, sea ice changes, differences among models and coupled process involved. Follow, we do the same analysis for the southern high latitudes and assess the reasons for asymmetries between poles.

## 3.1 Polar Amplification

In order to evaluate the seasonality of near surface polar warming, the seasons are defined as follows: December to February (DJF) as boreal winter, March to May (MAM) as boreal spring, June to August (JJA) as boreal summer, and September to November (SON) as boreal fall.

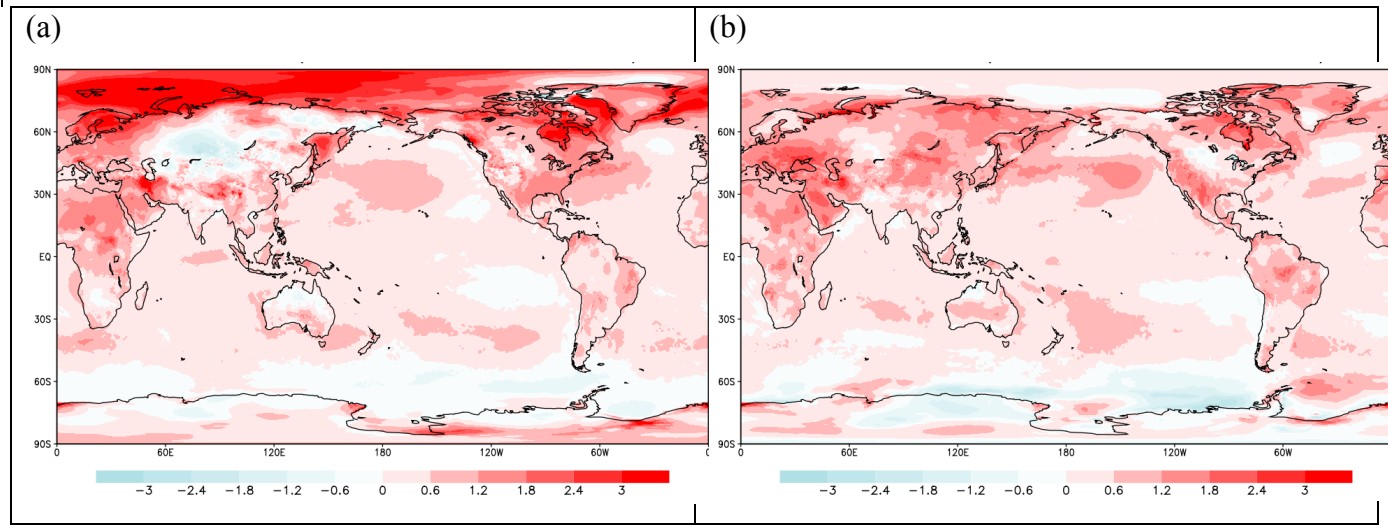

Figure 1. Polar Amplification using Long-term observations of Surface Air Temperatures ($^{o}$C) at 2008-2018 (seasonal average) relative to 1979-1989 (seasonal average) in (a) Winter (DJF) and (b) Summer (JJA). Source: Era Interim Reanalysis.

Figure 1 shows the enhanced surface warming at high latitudes compared to the rest of globe, with a slightly greater rate of warming in the 20[th] century. This Polar Amplification is not symmetric, most evidence is from Arctic region (during the boreal winter). According to Stocker et al., (2013), the enhanced warming at northern high latitudes was linked with decrease in snow cover and sea ice concentration, sea level rise and increase in land precipitation. Furthermore, changes in atmospheric and

ocean circulations (Chylek et al., 2019; Pedersen et al., 2016; Pithan and Mauritsen, 2014; Stocker et al., 2013; Yang et al., 2010; Graversen et al., 2008). Polar Amplification is also reported by Climate Models, driven by solar or natural carbon cycle perturbations (Sundqvist et al., 2010; O'ishi and Abe-Ouchi, 2011; Mann et al., 2009; Masson-Delmotte et al., 2006).

      Figure 2 shows the seasonality of the Polar Amplification (change in zonally SAT average)

simulated by BESM-AO V2.5 and 32 state-of-art CMIP5 and CMIP6 models. To assess the climate sensitivity of polar amplification, seasonally and coupled processes involved, we used the difference between Abrupt 4xCO$_2$ and piControl numerical experiments, considering only the last 30 years of the 150 years of model integration after quadrupling CO$_2$ concentration (when the model reaches a new equilibrium state). This procedure has been largely used by researchers since it allows us to evaluate

and compare potential warming and sensitivities among low and high latitudes as well as to compare differences between models (Van der Linden et al., 2019; Cvijanovic et al., 2015; Manabe et al., 2004; Holand and Bitz, 2003).

      Under the largest future GHG forcing (4xCO$_2$), the polar regions are found to be the most sensitive areas of the globe, with a very pronounced seasonality (Figure 2). The high southern latitudes

warming predicted by the models analyzed is modest in relation to the Arctic's, but still not negligible. This asymmetry is partly due to the smaller area covered by ocean in Northern Hemisphere that induces a smaller thermal inertia. Contrasting to the high latitudes, the tropical warming is similar for both hemispheres, without the robust warming pattern as showed in high latitudes. Salzmann (2017)

suggested that the overall weaker warming in Antarctica is due to a more efficient ocean heat uptake in the southern ocean, weaker surface albedo feedback in combination with ozone depletion. BESM-OA V2.5 model has no ozone chemistry as a climate component, so we suggest that even neglecting the ozone depletion, the weaker warming in Antarctica will be shown. Also is expected a weak albedo sea ice feedback compared with Arctic region (because the fast retreat of sea ice on the Northern Hemisphere). The role of the Antarctica surface height for both feedbacks processes and meridional transports is similarly important to consider. According to Salzmann (2017), the polar amplification asymmetry is explained by the difference in surface height. If Antarctica is considered to be flat in a climate simulation with $CO_2$-doubling experiment, the north-south asymmetry is reduced.

From September to February (boreal autumn and winter), the surface warming is maximum at northern high latitudes, decreasing with latitude to reaching a minimum at $60^o$S and then increasing towards the South Pole. Consistent with previous analyses based on climate simulations and observations, this enhanced Arctic Amplification appears as an inherent characteristic for the Arctic region (Pithan and Mauritsen, 2014). From March to August, the reverse signal shows the maximum warming close to $70^o$S, decreasing towards to tropical region, and lacking the enhanced warming at the northern high latitudes.

The main reason for winter (DJF) Arctic Amplification pointed by Serreze et al., (2009) is largely driven by changes in sea ice, allowing for intense heat transfers from the ocean to the atmosphere. During boreal summer, when Arctic warming is not prominent and solar radiation is maximal, the energy is used to melt sea ice and increase the sensible heat content of the upper ocean. The atmosphere heats the ocean during summer whereas the flux of heat is reverse in winter. The sea ice loss in summer allows a large warming of the upper ocean but the atmospheric warming at the surface or lower troposphere is modest (promoting more open water). The excess heat stored in the upper ocean is subsequently released to the atmosphere during winter (Serreze et al., 2009). According to Lu and Cai, (2009), in summertime the positive surface albedo feedback is mainly canceled out by the negative cloud radiative forcing feedback. The positive surface albedo feedback is relatively much weaker in winter when compared to its counterpart in summer, therefore it does not contribute to the pronounced polar amplification in winter.

For southern high latitudes, a pronounced warming appears from March to August (boreal summer and spring), predominantly close from 70°S. This enhanced warming tends to decrease in the direction of the South Pole. This pattern is similar to the one obtained by Goosse and Renssen, (2001). The authors used a coupled climate model to investigate the response of the Southern Ocean to an increase in GHG concentration. They found that the response could occur separatedly in two distinct phases. At the first moment, the ocean damps the surface warming (because of its large heat capacity). Then, after 100 years of run simulation, the warming is enhanced due to a positive feedback that is linked to a stronger oceanic meridional heat transport toward the southern ocean.

When comparing the seasonal response to $CO_2$ forcing among CMIP5 and CMIP6 models, for boreal winter (DJF), the enhanced Arctic warming at 75-90° N is shown to be a robust feature of all CMIP5 and CMIP6 climate models simulations presented here. For high Northern Hemisphere (high southern Hemisphere) the warming  (difference between piControl and $4xC0_2$) ranged from 10 K to 39 K (-0.5 K – 13 K). CAMS-CSM1-0 (CMIP6) and INMC-CM4 (CMIP5) presented the lowest warming, close from 12 K for Northern high latitudes. On the other hand, IPSL-CM6A-LR (CMIP6), HadGEM2-ES (CMIP5) and CanESM5 (CMIP6) outputs presented warming almost twice as large, with a high amplification close from 30 K. BESM model, for winter (DJF) season, presented Polar Amplification for Northern high latitudes, close from 27 K.

One interesting feature shown in Figure 2 is related to the maximum Arctic warming obtained in different simulations. Many models have shown that the maximum warming does not always occur at highest northern latitudes instead, it occurs between 80° N - 85° N decreasing toward 90° N. According to Holland and Bitz, (2003) the localization of the maximum warming varies widely among CMIP outputs, but models with high polar amplification generally presented a maximum warming over the Arctic Basin. Therefore, we suggest that the spatial distribution of maximum Arctic Amplification can be closely related to sea ice conditions though a sea ice albedo feedback, and this region (Arctic Basin) presents the major taxes of decrease in sea ice concentration. Similar result was found for the sea ice simulation from BESM model, as discussed below (Figure 4 and Table 1).  Additionally, Casagrande et al., (2016), using BESM-OA V2.3 model, showed that the sea ice spatial pattern could vary largely between CMIP5 models, especially in frontiers areas.

For the southern high latitudes, in wintertime (DJF- Figure 2), the warming decreases to close to $60^{o}$ S for most CMIP5 and CMIP6 models, increasing toward South Pole, with the maximum warming close to 11 K. The minimum warming is registered by GFDL-ESM2M and GFDL-ESM2G, both from CMIP5 simulations (close to 0K in $60^{o}$ S) and the maximum south polar amplification between models is presented by E3SM-1, close to $90^{o}$ S.

In summer (JJA), the compared response to $CO_2$ forcing in CMIP5 models is amplified (damped) at southern (northern) hemisphere. A pronounced amplification was found close to $70^{o}$S with a range of 1 K to 17K, decreasing towards the South Pole. In this region the maximum was obtained by BESM-OA V2.5 model, close to 13K.

       The pronounced seasonality of near surface warming in Polar Regions has been found in
observations (Bekryaev et al., 2010) and climate simulations (Holland and Bitz, 2003), but less emphasis has been placed in the vertical structure of the atmosphere. To understand if this enhanced warming occurs only in surface or also well above, Figure 3 presents results obtained with three different CMIP5 models with moderate (BESM-OA V2.5/MPI-ESM-LR) and low (NCAR-CCSM4) Polar Amplification (based on Figure 2).

Figure 3 shows evidence of temperature amplification well above the surface with enhanced warming during the cold season for both, northern and southern high latitudes. Snow and ice feedback cannot explain the warming above the lowermost part of the atmosphere because this feedback is expected to primarily affect the air temperature near surface. Part of the vertical warming may be explained by physical mechanisms that induce to warming as changes in the atmospheric heat transport
into the Arctic. According to Graversen et al., (2008), a substantial proportion of the vertical warming can be caused by changes in this variable, especially in summertime (JJA). Graversen and Wang (2009) used an idealized numerical experiment (doubling $CO_2$) with a climate model that has no ice albedo feedback. Their results also reveled a polar warming as a response to anthropogenic forcing (doubling $CO_2$). It was found that the enhanced Arctic warming is due to an increase of the atmospheric northward
transport of heat and moisture. These results are supported by observational analyses (Graversen et al., 2014; Graversen et al., 2006). In addition to ice-albedo feedback, the strength of the atmospheric stratification is an important factor to explain the vertical warming. The troposphere is more stably stratified in high latitudes. An increase in GHG forcing generates an increase in downwelling long-wave radiation at the surface, consequently causing warming, which in Polar Regions is confined to the lower
troposphere.

When examining Arctic warming at different levels computed by the three different models shown in Figure 3, we find that MPI-ESM-LR presented the strongest warming in both, near surface temperature and at high levels. Similar behavior is found at tropical regions, with robust warming at high levels (400-200 hPa). Holland and Bitz, (2003) suggested that sea ice conditions are more important than continental ice and snow cover to enhanced polar warming. According to these authors, models with relatively thin sea ice in the control run tend to have higher warming. The same feature was found in BESM-OA V2.5. According to Casagrande et al. (2016) and Casagrande (2016), the last version of BESM model (Version 2.5) is considered to be a climate model with high polar amplification exhibiting thin sea ice conditions on the control run. This occurs, in part, because of the new surface scheme based on Jimenez and Dudhia, (2012) and the microphysics of Ferrier et al. (2002). The advantage of these changes in the BESM´s last version is an improvement in the representation of precipitation, wind and humidity at tropical regions. Comparatively, NCAR-CCSM4 is considered a model with moderate polar amplification for both, Northern and Southern Ocean. The warming at high levels in boreal summer is not as amplified as in boreal winter. These results are in agreement with Holland and Bitz, (2003).

Figure 4 shows, under the largest future GHG ($4xCO_2$), the spatial pattern of sea ice changes for both, Arctic and Antarctic (difference between sea ice concentration for the last 30 years of abrupt4xCO2 numerical experiment and the last 30 years of the piControl run). The maximum of the Arctic warming obtained from observations (Figure 1) and different CMIP5 simulations (Figure 2) occurs in boreal winter (DJF).

According to Figure 2, the following models, in descending order, appear as having greater amplification: IPSL-CM6A-LR (CMIP6), HadGEM2-ES (CMIP5) and CanESM5 (CMIP6). Similar response, for the same period, is observed in Figure 4 and Figure 5, related to sea ice changes. Figure 5 shows the climatology of maximum and minimum sea ice area for the last 30 years of the abrupt $4xCO_2$ numerical experiment minus the last 30 years of the piControl run. For the Arctic, in March, EC-Earth3-Veg (NCAR-CCSM4) shows the highest (lowest) value, close to 15 x $10^6$ km$^{-2}$ (3 x $10^6$ km$^{-2}$). For September month, in agreement with Figure 2, the Polar Amplification is not evident as in the cold period. For the Antarctica (Figure 5), in the cold period (September), the difference between abrupt $4xCO_2$ numerical experiment and the piControl run is higher for models with enhanced Polar Amplification, as FGOALS-S2 (13 x $10^6$ km$^{-2}$). Both, Figure 4, Figure 5 and Table 1 are in agreement

with Figure 2, showing that the large decrease in sea ice concentration is more evident in models with great Polar Amplification, and for the same range of latitude ($75^o$ N – $90^o$ N). The end of melting period (when sea ice reaches its minimum annual value) for all models shows sea ice-free conditions (Table 1). Models that have strong Polar Amplification, also exhibit expressive changes in the annual amplitude of sea ice with outstanding ice-free condition from May to December (EC-Earth3-Veg) and June to December (HadGEM2-ES). Then, the end of melting period is expected early, likely, associated a large decrease in sea ice thickness, which contributes to a delay in sea ice formation. For BESM-OA V2.5, Arctic ice-free conditions are found from August to November. We suggest that, the Arctic will become covered only by first year sea ice (more vulnerable to melting), making the region more sensitive thermodynamically and dynamically to temperature changes. These evidences, corroborates with the theory, that the Arctic Polar Amplification is closely linked to sea ice albedo feedback.  For Antarctica, however, the same physical processes cannot be used to explain the Polar Amplification (as discussed previously). Although, according to Figure 2 and Figure 4, there is a small indication of the contribution of sea ice albedo feedback in Antarctic Polar Amplification, however, this still remain as a open discussion and we suggest that is important to consider the contribution of the ice sheet in Polar Amplification.

Previous researchers, using observational and modeling dataset, have found that shrinking of sea ice (Figure 4) and enhanced Arctic warming may affect the middle latitudes (Coumou et al., 2018; Screen, 2017; Walsh, 2014). According to Walsh (2014), the AA affects by the weakening the west-to-east wind speed in the upper atmosphere, by increasing the frequency of wintertime blocking events that in turn lead to persistence or slower propagation of anomalous temperature in middle latitudes, and by increasing in continental snow cover that can in turn influence the atmospheric circulation. Finally, in view of the results it is important to consider the limitations and differences among each climate model in order to improve the understanding of the physical process in climate simulations that represent large bias among the models belonging to CMIP5 project.

(a)

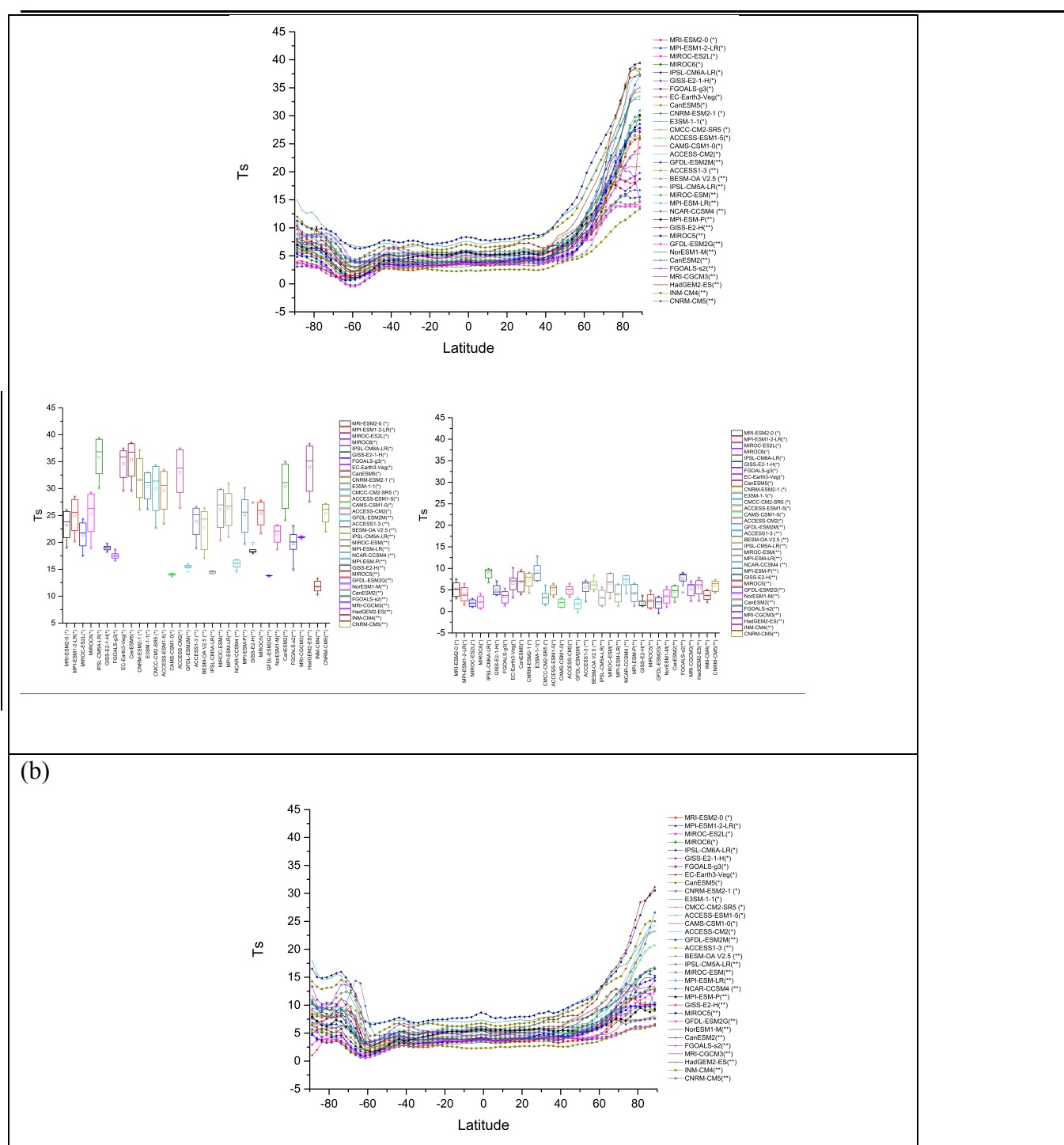

(b)

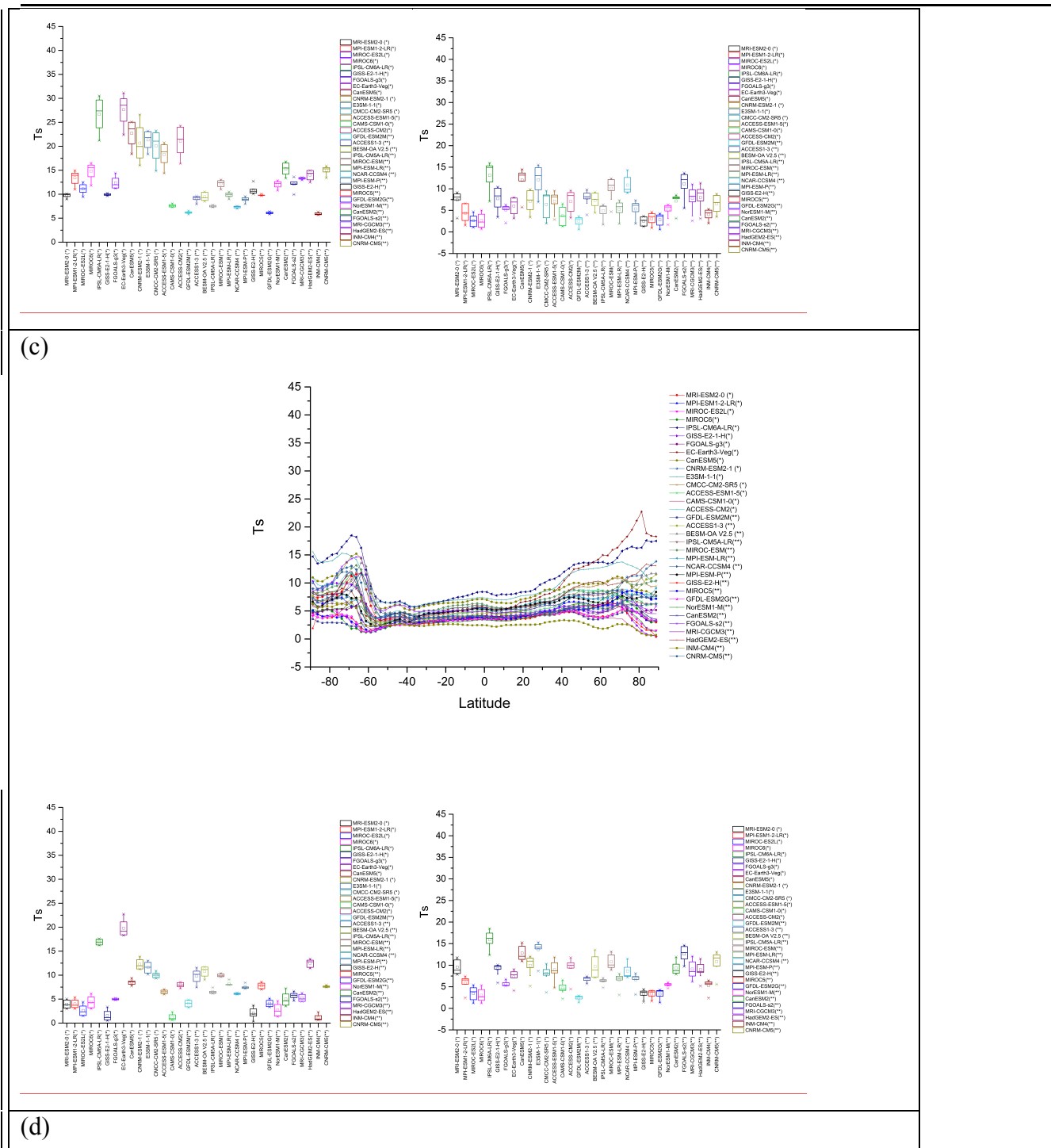

(c)

(d)

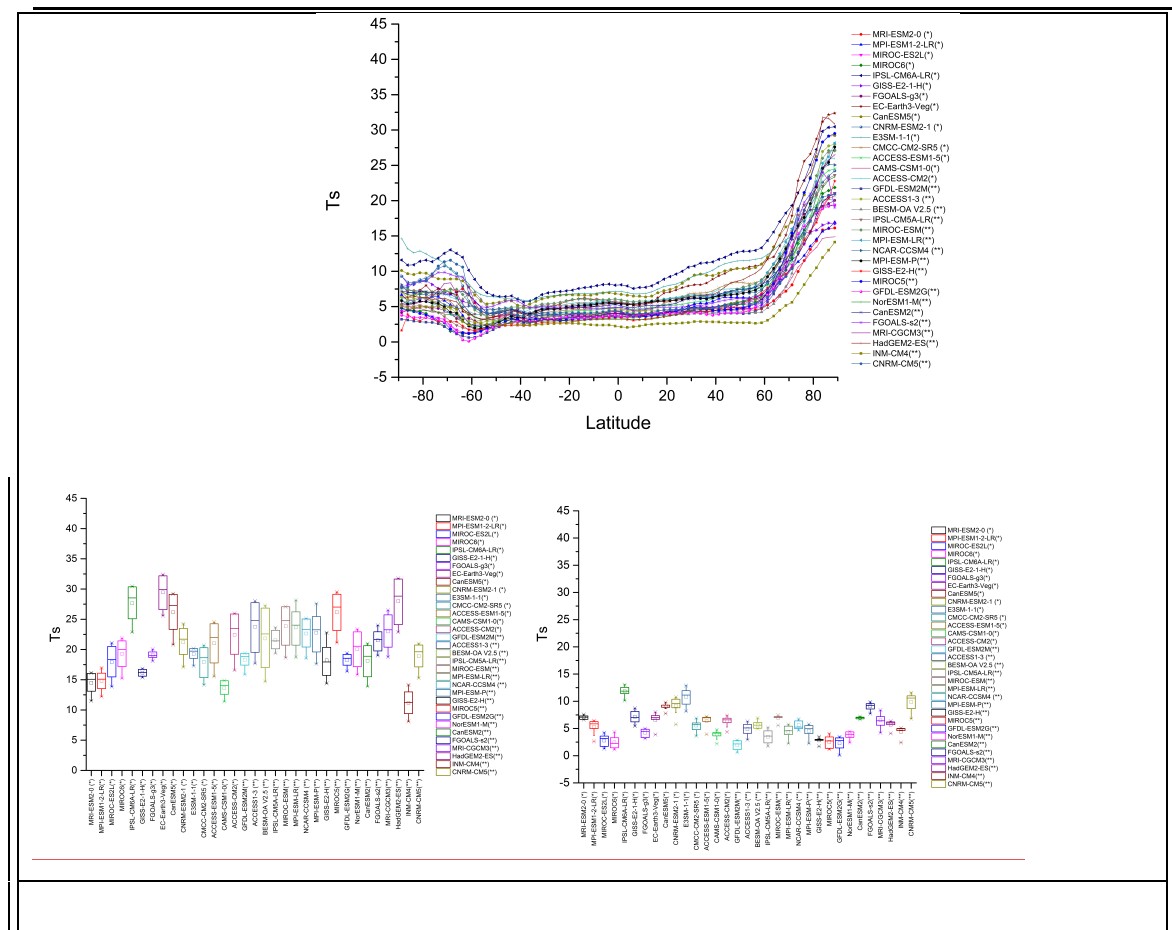

**Figure 2. Seasonal zonal mean surface temperature differences (K) for the last 30 years of Abrupt4xCO₂ numerical experiment minus the last 30 years of the piControl run for CMIP5 and CMIP6 models and box plot in 75°N – 90°N (left) and 60°S -80°S (right) for (a) Winter (DJF), (b) Spring (MAM), (c) Summer (JJA) and (d) autumn (SON).**

(a)

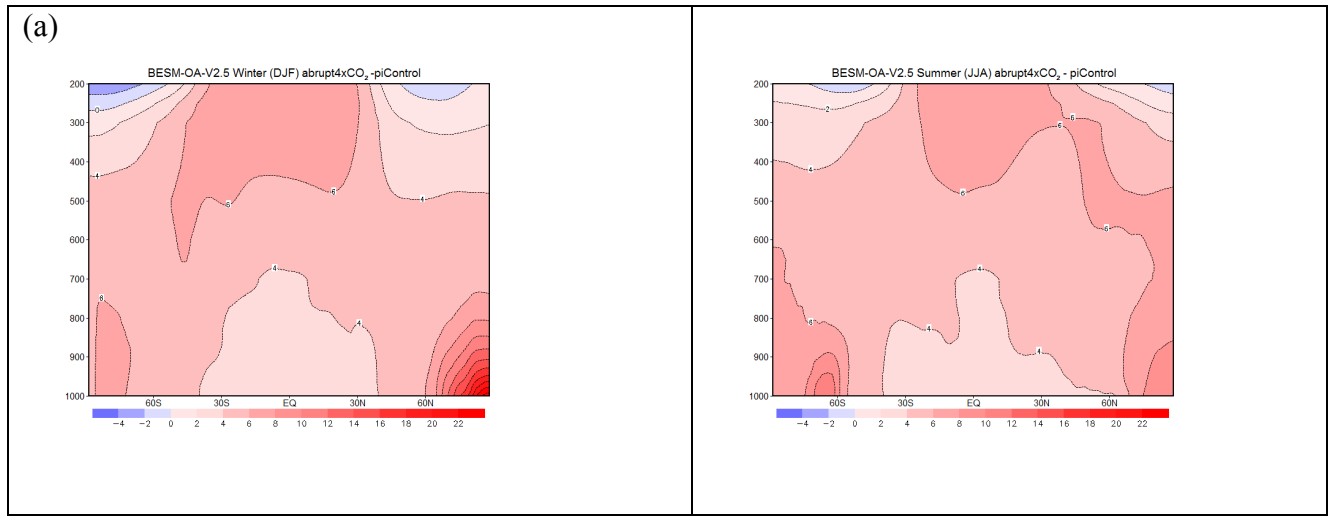

(b)

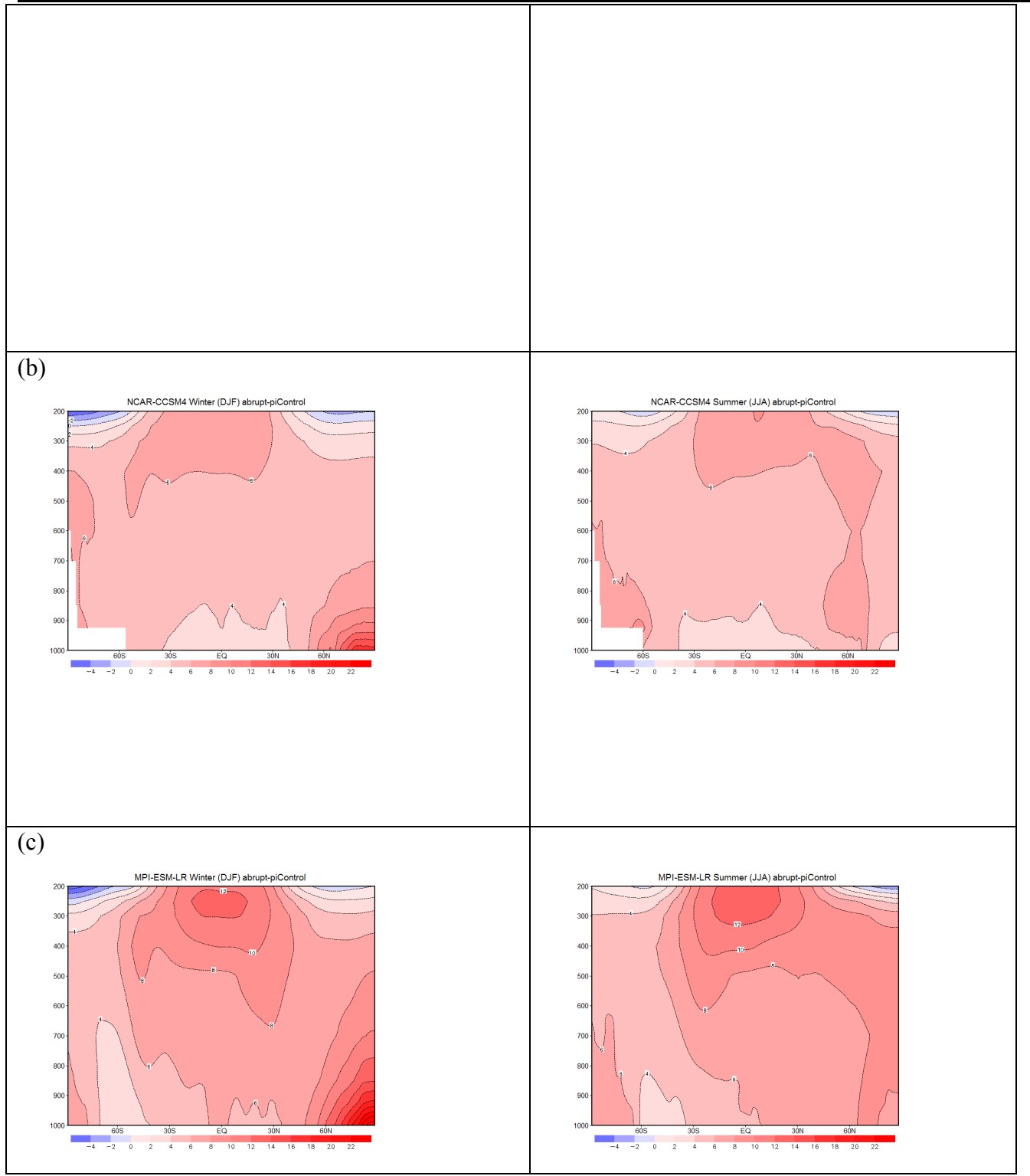

(c)

**Figure 3. Zonal-average atmosphere temperature changes, in $^{o}$C (Abrupt 4xCO$_2$ minus piControl) at each pressure level, in mb (solid line) for the last 30 years run for (a) BESM OA V2.5, (b) NCAR-CCSM4 and (c) MPI-ESM-LR model, in DJF (left) and JJA (right) columns.**

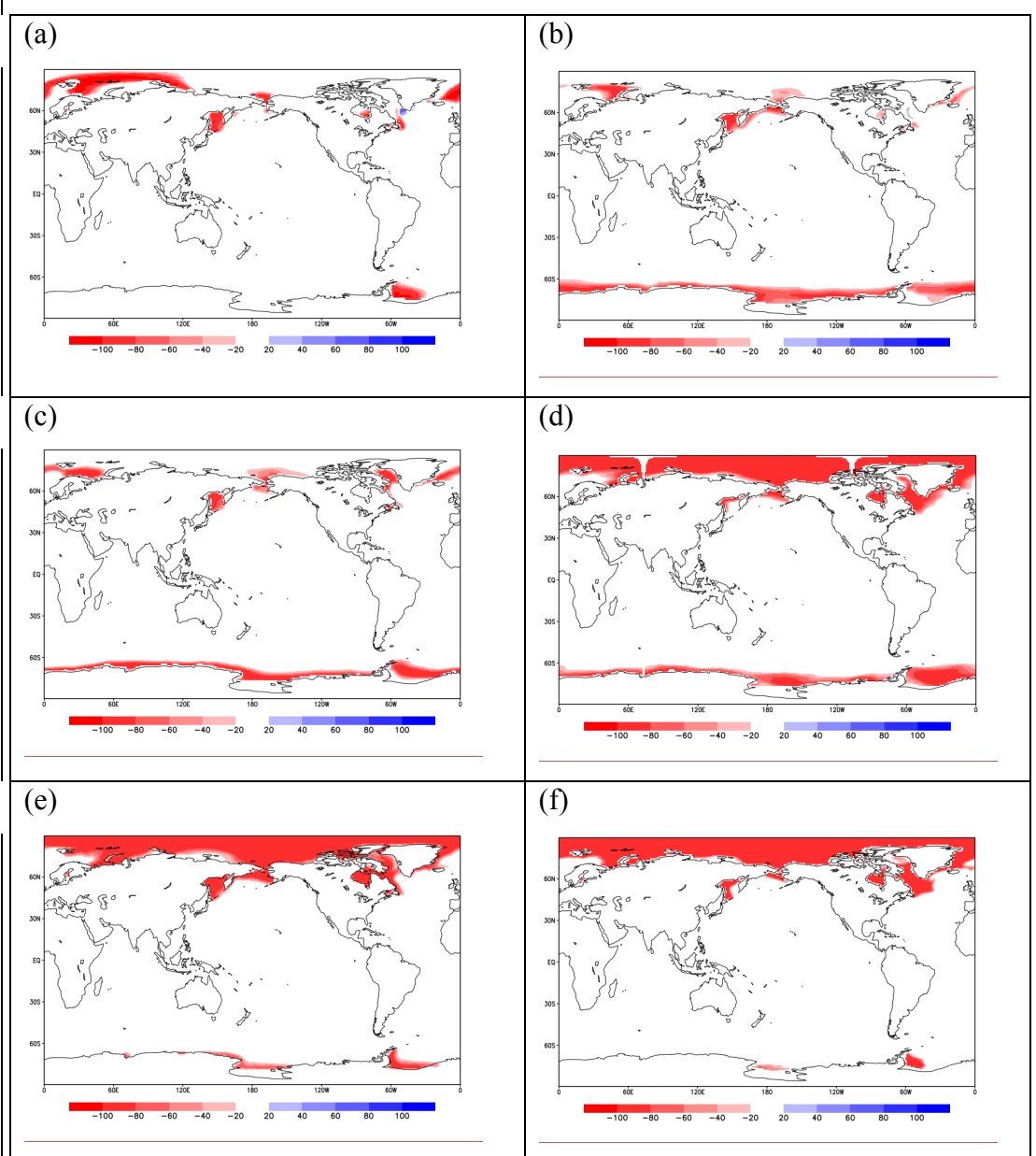

**Figure 4. Sea ice concentration for the last 30 years of abrupt4xCO₂ numerical experiment minus the last 30 years of the piControl run for the following models: (a) BESM-OA V2.5, (b) NCAR-CCSM4, (c) FGOALS-S2, (d) CanESM5, (e) HadGEM2-ES (f) EC-Earth3-Veg in March (left column).**

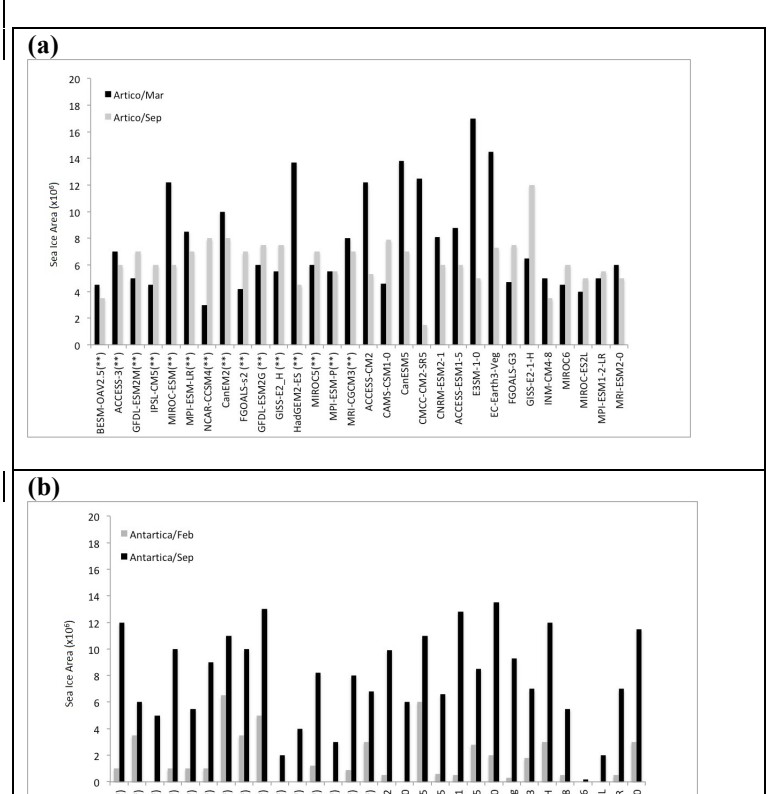

**Figure 5. Climatology of maximum and minimum Sea ice area (million square kilometers) for the last 30 years of the abrupt 4xCO₂ numerical experiment minus the last 30 years of the piControl run for CMIP5 and CMIP6 models (a) Arctic, black colors represents the March month and Gray colors represent September month (b) Antarctic: Black colors represents the September month and Gray colors represent February month.**

| CMIP6 Models | Arctic | March | Sept | Antarctic | Feb | Sept | CMIP5 Models | Arctic | March | Sept | Antarctic | Feb | Sept |
|---|---|---|---|---|---|---|---|---|---|---|---|---|---|
| | | March | Sept | | Feb | Sept | | | March | Sept | | Feb | Sept |
| ACCESS-CM2 | PiControl | 14,2 | 5,3 | PiControl | 0,5 | 13,4 | BESM-OAV2.5 | PiControl | 16 | 3.5 | PiControl | 1 | 29 |
| | 4xCO2 | 2 | Ice-Free [Jun-Dec] | 4xCO2 | Ice-Free [Jan-Mar] | 3,5 | | 4xCO2 | 11,5 | Ice-Free [Aug-Nov] | 4xCO2 | Ice-Free [Feb-Apr] | 17 |
| | | March | Sept | | Feb | Sept | | | March | Sept | | Feb | Sept |
| CAMS-CSM1-0 | PiControl | 17,1 | 7,9 | PiControl | Ice-Free | 12 | ACCESS-3 | PiControl | 14 | 6 | PiControl | 4,5 | 17 |
| | 4xCO2 | 12,5 | Ice-Free [Jul-Nov] | 4xCO2 | Ice -Free [Jan_Mar] | 6 | | 4xCO2 | 7 | Ice-Free [Jul-Nov] | 4xCO2 | 1 | 11 |
| | | March | Sept | | Feb | Sept | | | March | Sept | | Feb | Sept |
| CanESM5 | PiControl | 15 | 7 | PiControl | 6 | 20 | GFDL –ESM2M | PiControl | 14 | 7 | PiControl | Ice-Free | 9 |
| | 4xCO2 | 1,2 | Ice-Free [May-Dec] | 4xCO2 | Ice-Free [Feb-Mar] Ice-Free | 9 | | 4xCO2 | 9 | Ice-Free [Aug-Sep] | 4xCO2 | Ice-Free [Feb-Mar] | 4 |
| | | March | Sept | | Feb | Sept | | | March | Sept | | Feb | Sept |
| CMCC-CM2-SR5 | PiControl | 13,5 | 1,5 | PiControl | 0,6 | 14 | IPSL –CM5-LRM | PiControl | 13 | 6 | PiControl | 1 | 17 |
| | 4xCO2 | 1 | Ice-Free [Marc-Dec] | 4xCO2 / 4xCO2 | Ice-Free [Jan-Mar] Ice-Free | 7,4 | | 4xCO2 | 8,5 | Ice-Free [Jul-Oct] | 4xCO2 | Ice -Free [Jan-Mar] | 7 |
| | | March | Sept | | Feb | Sept | | | March | Sept | | Feb | Sept |
| CNRM-ESM2-1 | PiControl | 15,2 | 6 | PiControl | 0,5 | 16 | MIROC-ESM | PiControl | 13 | 6 | PiControl | 1 | 14 |
| | 4xCO2 | 7,1 | Ice-Free [Jul-Dec] | 4xCO2 | Ice free (Jan-Apr) | 3,2 | | 4xCO2 | 0,8 | Ice-Free [May-Dec] | 4xCO2 | Ice free | 8.5 |
| | | March | Sept | | Feb | Sep | | | March | Sept | | Feb | Sep |
| ACCESS-ESM1-5 | PiControl | 13,5 | 6 | PiControl | 2,8 | 14 | MPI –ESM-LR | PiControl | 12 | 7 | PiControl | 1 | 13 |
| | 4xCO2 | 4,7 | Ice-Free [Jul-Dec] | 4xCO2 | Ice-Free [Feb-Mar] | 5,5 | | 4xCO2 | 3,5 | Ice-Free [Jun-Dec] | 4xCO2 | Ice-Free [Jan-Apr] | 4 |
| | | March | Sept | | Feb | Sept | | | March | Sept | | Feb | Sept |
| E3SM-1-0 | PiControl | 17 | 5 | PiControl | 2 | 17 | NCAR – CCSM4 | PiControl | 13 | 8 | PiControl | 7.5 | 22 |
| | 4xCO2 | Ice-Free | Ice-Free [All Year] | 4xCO2 | Ice-Free | 3,5 | | 4xCO2 | 10 | Ice-Free [Aug-Oct] | 4xCO2 | 1 | 11 |
| | | | Sept | | Feb | Sept | | | March | Sept | | Feb | Sept |
| EC-Earth3-Veg | PiControl | 15 | 7,3 | PiControl | 0,3 | 10,5 | CanESM2 | PiControl | 15 | 3,8 | PiControl | 4 | 22 |
| | 4xCO2 | 0,5 | Ice-Free [May-Dec] | 4xCO2 | Ice-Free | 1,2 | | 4xCO2 | 5 | Ice-Free [Jul-Nov] | 4xCO2 | 0,.5 | 12 |
| | | March | Sept | | Feb | Sept | | | March | Sept | | Feb | Sept |
| FGOALS-G3 | PiControl | 15 | 8 | PiControl | 2,8 | 19 | FGOALS-s2 | PiControl | 12 | 7 | PiControl | 6 | 22 |
| | 4xCO2 | 10,3 | 0,5 | 4xCO2 | 1 | 12 | | 4xCO2 | 7,8 | Ice-Free [Ago-Out] | 4xCO2 | 1 | 9 |
| | | March | Sept | | Feb | Sept | | | March | Sept | | Feb | Sept |
| GISS-E2-1-H | PiControl | 22 | 12 | PiControl | 3,5 | 21 | GFDL-ESM2G | PiControl | 18 | 7,5 | PiControl | Ice-Free | 11,5 |
| | 4xCO2 | 15,5 | Ice-Free [Aug-Sep] | 4xCO2 | 0,5 Ice-Free | 9 | | 4xCO2 | 12 | Ice-Free [Aug-Sep] | 4xCO2 | Ice-Free [Jan-Mar] Ice-Free | 9,5 |
| | | March | Sept | | Feb | Sept | | | March | Sept | | Feb | Sept |
| INM-CM4-8 | PiControl | 14 | 7 | PiControl | 10 | | GISS-E2_H | PiControl | 15 | 7,5 | PiControl | Ice-Free | 7,5 |
| | 4xCO2 | 9 | 3,5 | 4xCO2 / 4xCO2 | Ice-Free [Jan-Mar] Ice-Free | 4,5 | | 4xCO2 | 9,5 | Ice-Free [Aug-Oct] | 4xCO2 / 4xCO2 | Ice-Free [Jan-Mar] Ice-Free | 3,5 |
| | | March | Sept | | Feb | Sept | | | March | Sept | | Feb | Sept |
| MIROC6 | PiControl | 11 | 6 | PiControl | Ice-Free | 3 | HadGEM2-ES | PiControl | 16 | 4,5 | PiControl | 1,2 | 14,2 |
| | 4xCO2 | 6,5 | Ice-Free [Jul-Dec] | 4xCO2 | Ice free | 2,8 | | 4xCO2 | 2,3 | Ice-Free [Jun-Dec] | 4xCO2 | Ice free | 6 |
| | | March | Sept | | Feb | Sep | | | March | Sept | | Feb | Sep |
| MIROC-ES2L | PiControl | 12 | 5 | PiControl | Ice-Free | 3 | MIROC5 | PiControl | 13 | 7 | PiControl | Ice-Free | 7 |
| | 4xCO2 | 8 | Ice-Free [Jul-Nov] | 4xCO2 | Ice-Free [Jan-Mar] | 1 | | 4xCO2 | 7 | Ice-Free [Jul-Nov] | 4xCO2 | Ice-Free [Jan-Mar] | 4 |
| | | March | Sept | | Feb | Sep | | | March | Sept | | Feb | Sep |
| MPI-ESM1-2-LR | PiControl | 12 | 5,5 | PiControl | Ice-Free | 11 | MPI-ESM-P | PiControl | 12 | 5,5 | PiControl | 0,9 | 12 |
| | 4xCO2 | 7 | Ice-Free [Jul-Nov] | 4xCO2 | Ice-Free [Jan-Mar] | 4 / 3 | | 4xCO2 | 6,5 | Ice-Free [Jul-Dez] | 4xCO2 | Ice-Free [Jan-Mar] | 4 |
| | | March | Sept | | Feb | Sep | | | March | Sept | | Feb | Sep |
| MRI-ESM2-0 | PiControl | 15 | 5 | PiControl | 3 | 20,5 | MRI-CGCM3 | PiControl | 21 | 7 | PiControl | 3 | 19 |
| | 4xCO2 | 9 | Ice-Free [Jul-Dez] | 4xCO2 | Ice-Free [Fev] | 9 | | 4xCO2 | 13 | Ice-Free [Jul-Oct] | 4xCO2 | Ice-Free [Fev] | 12,2 |

**Table 1. Climatology of maximum and minimum Sea ice area (million square kilometers) for the last 30 years of the abrupt 4xCO₂ numerical experiment and the last 30 years of the piControl run for CMIP5 and CMIP6 models.**

## 4 Conclusion

Polar amplification is possibly one of the most important sensitive indicators of climate change. Robust patterns of near-surface temperature response to global warming at high latitudes have been identified in recent studies (Smith et al., 2019; Stuecker et al., 2018; Pithan and Mauritsen, 2014). For northern high latitudes, the shrinkage of sea ice as response to increase of GHC is one of the most cited reasons (Serreze and Barry, 2011; Kumar et al., 2010; Screen and Simmonds, 2010). Here we analyzed the seasonality of polar amplification using some CMIP5 coupled climate models in a quadrupling $CO_2$ numerical experiment for both, North and South Hemispheres. Our results showed that the Polar Regions are much more vulnerable to a large warming due to an increase in atmospheric $CO_2$ forcing, than the rest of the world, particularly during the cold season. For northern high latitudes, the feedback albedo-se` ice contributes to decrease in sea ice cover, exposing new expanses of ocean and land surfaces (leading to greater solar absorption), thus amplifying the accelerated warming and driving future melting. Despite of the asymmetry in warming between Arctic and Antarctic, both poles show systematically polar amplification in all climate models. Different physical processes acts to explain the sensibilities between poles. While in Northern high latitudes the warming is closely related to sea ice albedo feedback, in southern high latitudes the amplification is related to thermal inertia, combination of changes in winds and ozone depletion. We detected three climate models as having high amplification, in cold season_for Arctic: IPSL-CM6A-LR (CMIP6), HadGEM2-ES (CMIP5) and CanESM5 (CMIP6). For South Hemisphere, in the cold season (JJA), the Climate Model identified as having high polar amplification were: IPSL-CM6A-LR (CMIP6), CanESM5(CMIP6) and FGOALS-s2 (CMIP5). For high Northern Hemisphere (high southern Hemisphere) the warming ranged from 10 K to 39 K (-0.5 K – 13 K), INM-CM4 (CMIP5) presents the lowest warming, close from 10 K for Northern high latitudes. For Antarctica, the maximum warming, close to 14 K is presented by FGOALS-s2, close to $70^{o}$ S. The vertical profiles of air temperature showed stronger warming at the surface, particularly for northern high latitudes, indicating the effectiveness of the albedo-sea ice feedback. Furthermore, we evaluated the linkage between sea ice changes and Polar Amplification from different CMIP5 models. We found that large decreases in sea ice concentration are more evident in models with great Polar Amplification, and for the same range of latitude ($75^{o}$ N – $90^{o}$ N). We suggest, according to our results,

that the large difference between models might be related to sea ice initial conditions. Therefore, those differences are also related to the parameterizations used to represent changes in clouds and energy balance. The coupled ocean-atmosphere-cryosphere physical processes involved in high-latitudes climate changes are fully inter-dependent with complicated structures contending with each other at many temporal and spatial scales. Until now, the complexities of the multiple coupled processes lead to

a leak in reproducibility by the numerical climate models, especially at southern regions. The sparse and short data record does not help also. Nevertheless, even with inherent limitations and uncertainties, the Global Climate Models are the most powerful tools available for simulating the climatic response to GHG forcing and to providing future scenarios to community.

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
