# Peer review of "Inter-hemispheric seasonal comparison of Polar Amplification using radiative forcing of quadrupling CO2 experiment"

_Annales Geophysicae, 2019_

## Referee Comment (RC1) · Anonymous Referee #1 · 9 Oct 2019

The authors analyzed the seasonality of polar amplification using CMIP5 coupled climate models and BESM to compare 4xCO2 and piControl experiments. The results show that there is an asymmetry in the polar amplifications between the two hemispheres. I do believe that the scientific subject and aims of this paper are within the scope of ANGEO and this paper is easy to read, but I also think that this specific paper could need a significant amount of new analysis and modeling work in order to appropriately address its hypotheses.

Major comments:

I am not sure whether this paper is a letter or an article. If it is a letter paper, I didn't see

relatively interesting points worthy of publication on ANGEO in the present version. If it is an article paper, I think it should include more analysis results. For example, it will be better to show the asymmetry in the seasonality of polar amplification in different hemispheres in observations. The authors cited many references and used a lot of 'suggest' to explain the figures. However, I think it may be better to give a direct evidence and make more figures to support them, e.g., L197 sea ice patterns and L237 plotting atmospheric heat transport in different models, etc.

Specific comments:

L40: You mentioned 'numerous scientific publications', but there is only one reference in the end of this sentence.

L42: suggest->suggested

L63: Please give explanations regarding why the performance of Arctic simulation is better.

L75: are also depended on-> depend on

L76: making -> which makes

L105: last -> latest

L158: 'with no so enhanced warming' is confusing. Please rephrase it.

L163: looses heat to -> heats

L174-L178: Why the authors mentioned the linkage between Arctic sea ice loss and mid-latitude weather? I think it is irrelevant to your topic.

L197: It may be better to have a spatial distribution of sea ice trends to support your hypothesis, see my major comments.

L209: last->latest

L247: It is difficult for me to link Fig.3 to deep convection. I think the authors should

give more evidences to support your conclusions. See my major comments.
* * *

---

## Referee Comment (RC2) · Anonymous Referee #2 · 18 Oct 2019

The authors studied the seasonality of the Polar Amplification (PA), here defined as the difference between the CMIP5 control (piControl) and the abrupt increased CO2 (abrupt4xCO2) experiments. The manuscript shows that the Arctic is more sensible to the PA, which is more remarkable in autumn and winter. I think this is a timely and interesting topic that certainly deserves attention from the scientific community. Also, the paper fits well in the short-communication format (I am assuming this is the case). In my opinion, the manuscript has potential, but the analyses presented so far are too shallow and it should be further improved before publication. First, I think the authors should use all the CMIP models available, and not only 6 of them: CMIP5 has 31 models that performed the abrupt4xCO2 experiment. If there is a good reason to

use only these 6 models, this should be acknowledged in the text. By using a larger number of models, the authors would be able to perform some statistical analysis (e.g., to compare BESM against the others) and bring robustness to the manuscript. Second, it would be much more useful to the scientific community to see this study conducted with the CMIP6 outputs. Again, the data processing and analyses performed so far are straightforward so that it should not be a problem to adapt them to the CMIP6 models. Third, although I understand that is fair to use the abrupt 4xCO2 experiments in this study, the authors could bring other experiments to their analyses such as the 1pctCO2. Fourth, as the first reviewer also pointed out, I also think that many of the hypotheses raised by the authors could be effectively tested with the CMIP data. Fifth, I miss in the introduction a strong point on what this manuscript brings as new results and, I also missed a more comprehensive conclusion for the new findings. Finally, a bit more care with manuscript writing is required. I have pointed out some mistakes below (not exhaustively), as well as other comments that could be considered by the authors for improving their manuscript.

Other comments

Pg. 1; L. 8: "The numerical climate simulation from Brazilian Earth System Model (BESM) are..." – Replace "are" by "is" or "simulation" by "simulations".

Pg. 1; Ls. 18, 19, 21, 24: Consider to add an article in the following cases – "warming at the surface", "heat in the atmosphere.", "for the cold season", and "in the coming decades". Also, for other instances in the manuscript.

Overall comment: For uncountable nouns, the use of the indefinite article "a" may be redundant. For instance: "a warming", "a cooling". This rule could be considered for the entire manuscript.

Pg. 2; Ls. 31: I guess the authors meant GHG rather "GHC".

Pg. 2; Ls. 35–39: The sentence is confusing. It is kind of hard to get what the authors

mean. Please, consider to rewrite it. For instance, "these two-poles inter-hemispheric asymmetries in the mean ocean circulation" but nothing was mentioned for the "Arctic mean circulation".

Pg. 2; L. 37: "According Marshall..." replace by "According to Marshall". Please, check for the other instances in the text.

Pg. 2; Ls. 40–42: "Numerous..." but only Vaughan was cited.

Pg. 2; Ls. 45–46: "from between 1875 and 2008" – Drop "from".

Pg. 2; Ls. 46–47: Add "the" in "latitudes of the northern hemisphere".

Pg. 2; L. 55: Replace "this processes" by "these processes"; Also, it seems that the explanation "Ocean is becoming more like the Atlantic ocean" is not required.

Pg. 2; L. 59: "The large differences among the models is" – Replace "is" by "are".

Pg. 3; Ls. 78–81: I was wondering why comparing the BESM results against only 5 other models rather than the entire ensemble of models? Also, since we are already in the CMIP6, why not make this study with experiments from this phase. In addition, since the 4xCO2 seems a bit unrealistic, I think the use of the simulations forced by "1% per year CO2 increase (1pctCO2; Eyring et al., 2016)" would strength the manuscript.

Pg. 3; L. 81: "The paper was is organized".

Pg. 3; L. 86: Missing "." at the end of the sentence.

Pg. 3; L. 93: "an a instantaneous"; "the 21st".

There is a mistake with numbering sections as per Sec. 3.

Pg. 5; L. 129: "accesses". Do you mean "assess"?

Pg. 5; L. 128–129: It does not seem to be the case since the discussion for Arctic and Antarctic is, in some instances, merged in Sec. 3.

Pg. 5; L. 135: Replace "assesses" by "assess".

Pg. 6; L. 138: Replace "This procedure been largely" by "This procedure has been". Also, the authors argued "largely" but cited only 2 references.

Pg. 6; L. 138: "Contrasting, the tropical warming for both, northern and southern hemisphere, is pretty similar with not so accentuated SAT increase in summer and for regions close to 30N." – Not sure I agree with this statement. From Fig. 1, it is noticeable an increase in the SAT differences from about -60S to +60N. Could the authors add some words/explanation for that in the manuscript?

Pg. 6; L. 146–147: "... the overall weaker warming in Antarctica is due to a more efficient ocean heat uptake in the southern ocean". I am wondering whether the authors could test this by looking at the SST data (or another output variable). For instance, is the Polar Amplification and respective seasonal cycle also observed in the SST data. If so, what are the differences between Antarctic and Arctic? Maybe something could be shown in terms of albedo feedback. I think this is a better way to address the issue rather than "We expect...".

Pg. 6; L. 155: "reaching a minimum at 70S" – I would rather say 60S.

Pg. 6; L. 160: "The main reason for winter (DJF) Arctic Amplification pointed by Serreze et al., (2009) is largely driven by changes in sea ice, allowing for intense heat transfers from the ocean to the atmosphere...". I also think the authors could check this hypothesis with the CMIP datasets.

Pg. 6; L. 163: Replace "looses" by "loses".

Pg. 7; L. 171: Replace "consequent" by "consequently".

Pg. 7; L. 174–178: The referred teleconnection seems to be out of context here.

Pg. 7; L. 180: Replace "trend" by "tends"(?)

Pg. 7; L. 190: Replace "In the other hand" by "On the other hand".

Pg. 7; L. 197: Replace "Artic" by "Arctic".

Pg. 7; L. 203: Replace "register" by "registered".

Pg. 8; L. 209: Replace "previously version" by "previous version".

Pg. 8; L. 208–212: Not sure the comparison between the two BESM versions makes sense in the scope of the manuscript. The paper compares different models but not different versions of the same model. As it is, it seems like an artifact for auto-citation.

Fig. 2 – I think this analysis should be performed for the ensemble of models.

Fig. 3: This figure should be further improved. The labels are too small; it is missing the y-label and unity; the colorbar is not aligned with the figures.

Pg. 11; L. 275: Replace "a combination changes in winds" by "a combination of changes in winds"

---

## Referee Comment (RC4) · Anonymous Referee #3 · 27 Oct 2019

Major Comments: The authors present here the seasonality of polar amplification (PA) defined as the difference between the different numerical models. I believe the article has a lot of potential because its results show the importance of these analyzes for these regions, and also a greater approach on the subject. However, the article should be enhanced for future publication. For example, the objective is not clear in the Summary. There is no more detailed description of the main objectives that will be addressed in the work. Although the objective is described in the introduction, I find it necessary to present this objective also in the abstract of the article. In addition, the

results presented also require a more refined discussion. More details, more comparisons are needed for the new version. The conclusions also need to be improved by showing the importance of the work, a well-explanatory summary of the results ...

Specific Comments:

Page 2, L. 40: "Numerous Scientific Publications"? I suggest rewriting this paragraph because it is confusing.

Page 2, L. 56: References ..?

Page 5, L. 129: Replaced "parsed" with "parsed"

Page, L. 132: Attention to section description: 3.1 Polar ...
* * *

---

## Author Comment (AC1) · 10 Feb 2020

Casagrande et al. (2019).

Response to Referee #1

Answer:

Thank you very much for your consideration. We really appreciate the comments and have learned a lot. Appropriate changes were made in the revised manuscript according to the suggestions.

First, this paper is an article. We added results as you suggest, for example: analysis

[Figure]

of polar amplification from observational data (Figure 1) and sea ice analysis from different CMIP5 models (Figure 3, Table 1 and Figure 4). This analysis provided greater robustness in the results, which were included here in several parts of the revised manuscript. Thus replacing, expressions as "we suggest" with more complete discussions.

Figure 1. Polar Amplification using Long-term observations of Surface Air Temperatures (oC) at 2008-2018 (seasonal average) relative to 1979 -1989 (seasonal average) in (a) Winter (DJF) and (b) Summer (JJA). Source: Era Interim Reanalysis.

Figure 1 shows the enhanced surface warming at high latitudes compared to the rest of globe, with a slightly greater rate of warming in the 20th century. The observed Polar Amplification is not symmetric, most evidence is from Arctic region (during the boreal winter). According to Stocker et al., (2013), the enhanced warming at northern high latitudes was linked with decrease in snow cover and sea ice concentration, sea level rise and increase in land precipitation. Besides that, changes in atmospheric and ocean circulation (Chylek et al., 2019; Pedersen et al., 2016; Pithan and Mauritsen, 2014; Stocker et al., 2013; Yang et al., 2010; Graversen et al., 2008).

Following the reviewer's suggestion and in order to better discuss the relationship between enhanced warming at high latitudes (Figure 1) and sea ice changes, we include the Figure 3, Figure 4 and Table1.

Figure 3 (new - attached here) shows, under the largest future GHG (4xCO2), the spatial pattern of sea ice changes for both, Arctic and Antarctic (difference between sea ice concentration for the last 30 years of abrupt4xCO2 numerical experiment and the last 30 years of the piControl run). This new Figure complements and makes the discussion shown in Figure 1 (old manuscript) more robust. The maximum of the Arctic warming obtained from observations (new Figure 1) and different CMIP5 simulations (old Figure 1) occurs in boreal winter (DJF). According to Figure 1 (old manuscript), the following models, in descending order, appears as having greater amplification:

MIROC – ESM, MPI-ESM, BESM-OA V2.5 and CSIRO-ACCESS. Similar response, for the same period is observed in Figure 3 and Figure 4, related to sea ice changes. The large decrease in sea ice concentration is more evident in models with great Polar Amplification, and for the same range of latitude (75o N – 90o N). The end of melting period (when sea ice reaches its minimum annual value) for all models shows sea ice-free conditions. Models that have strong Polar Amplification exhibit expressive changes in the sea ice annual amplitude with outstanding ice-free condition from may to December (MIROC-ESM) and June to December (MPI-ESM). Then, the end of melting period is expected early, likely, associated a large decrease in sea ice thickness and contributing to a delay in sea ice formation. We suggest, based in Figure 3 and Table 1, that, the Arctic will become covered only by first year sea ice (more vulnerable to melting), making the region more sensitive thermodynamically and dynamically to temperature changes. These new evidences presented here, corroborates with the theory, that the Polar Amplification is closely linked to sea ice albedo feedback. For Antarctica, however, the same physical processes cannot be used to explain the Polar Amplification (as discussed in the manuscript). Although, according to Figure 1 (old manuscript) and Figure 3 (new - attached here), there is a small indication of the contribution of sea ice albedo feedback in Antarctic Polar Amplification. Latitudes between 60oN and 65oN (greater Polar Amplification, models BESM-OAV2.5, MIROC-ESM and NCAR-CCSM4) for Austral winter also have trace of relation with abrupt changes in sea ice (Figure 3). Here, it is important to consider the contribution of the ice sheet in Polar Amplification that is not represented by the most of CMIP5 current models. According to Salzmann (2017 the overall weaker warming in Antarctica is due to a more efficient ocean heat uptake in the southern ocean, weaker surface albedo feedback in combination with ozone depletion.

Figure 3. Sea ice concentration for the last 30 years of Abrupt4xCO2 numerical experiment minus the last 30 years of the piControl run for the following models: BESM-OA V2.5, NCAR-CCSM4, GFDL-ESM-LR, MPI-ESM-LR, CSIRO, IPSL and MIROC-ESM in March (left column) and September (right column).

Table 1. Sea ice area (million square kilometers) for the last 30 years of the abrupt 4xCO2 numerical experiment minus the last 30 years of the piControl run for the following models: BESM-OA V2.5, NCAR-CCSM4, GFDL-ESM-LR, MPI-ESM-LR, CSIRO, IPSL and MIROC-ESM. I Arctic (Antarctic) sea ice reach its annual maximum area in march (february) and the minimum period in September.

Figure 4. Climatology of maximum and minimum Sea ice area (million square kilometers) for the last 30 years of the abrupt 4xCO2 numerical experiment minus the last 30 years of the piControl run for the following models: BESM-OA V2.5, NCAR-CCSM4, GFDL-ESM-LR, MPI-ESM-LR, CSIRO, IPSL and MIROC-ESM. (a) Arctic, (b) Antarctic. Black color represents the maximum (minimum) period of sea ice concentration, march (february) month for Arctic (Antarctic). Gray color bar represents September month.

Figure 5. Scatter plot of the SAT (K) and SIC (%) for the last 30 years of Abrupt4xCO2 numerical experiment minus the last 30 years of the piControl run for the following models: BESM-OA V2.5, NCAR-CCSM4, GFDL-ESM-LR, MPI-ESM-LR, CSIRO, IPSL and MIROC-ESM. The blue (black) dots represent march (september) month average, respectively the maximum and the minimum of sea ice concentration.

Specific comments: L40: You mentioned 'numerous scientific publications', but there is only one reference in the end of this sentence. Reply: ok Numerous scientific publications based on both, observations and state-of-the-art Global Climate Model simulations for the high latitudes of the northern hemisphere have shown that AA is an intrinsic feature of the Earth's climate system (Smith et al., 2019; Vaughan et al., 2013; Serreze and Barry, 2011; Screen and Simmonds, 2010).

L42: suggest->suggested Reply: ok

L63: Please give explanations regarding why the performance of Arctic simulation is better. Reply: ok According to Shu et al., (2015), Global Climate Models simulations in general offer much better simulations for the Arctic than for the Antarctica. Turner et

al., (2015) suggested that the main p roblem of climate models in the high latitudes of the southern hemisphere is their inability to reproduce the observed (although slight) increase in Sea Ice Extent (SIE). Bintanja et al., (2015) and Swart and Fyfe, (2013) have demonstrated the importance to include the effect of the increasing freshwater input from Antarctic continental ice into the Southern Ocean. The authors describe that the ice sheet dynamics, essential for having accurate sea ice simulations, is currently disregarded in all CMIP5 models. Swart and Fyfe (2013) also suggested that this deficiency may significantly influence the simulated sea ice trend because the subsurface ocean warming causes basal ice-shelf melt, freshening the surface waters, which eventually leads to an increase in sea ice formation.. Moreover, the instrumental network for data collection in Antarctica and the Southern Ocean is considered scarce (even more than in the Arctic), inhomogeneous and insufficiently dense to validate climate models. Therefore, or the high latitudes regions of the southern hemisphere, the effects of the ongoing climate change and its associated processes are still considered hot topics that lack conclusive answers.

L75: are also depended on-> depend on Reply: ok

L76: making -> which makes Reply: ok

L105: last -> latest Reply: ok

L158: 'with no so enhanced warming' is confusing. Please rephrase it. Reply: ok From March to August, the reverse signal shows the maximum warming close to 70oS, decreasing towards to tropical region, lacking the enhanced warming at the northern high latitudes.

L163: looses heat to -> heats Reply: ok

L174-L178: Why the authors mentioned the linkage between Arctic sea ice loss and mid-latitude weather? I think it is irrelevant to your topic. Reply: ok , it is out.

L197: It may be better to have a spatial distribution of sea ice trends to support your

hypothesis, see my major comments. Reply: ok , Figure 3 and Table 1.

L209: last->latest Reply: ok

L247: It is difficult for me to link Fig.3 to deep convection. I think the authors should give more evidences to support your conclusions. See my major comments.

Reply: ok, we agree with suggestion.

Winter DJF Obs (2008−2018 minus 1979−1989)

SummerJJA Obs (2008−2018 minus 1979−1989)

Figure 1. Polar Amplification using Long-term observations of Surface Air Temperatures ($^o$C) at 2008-2018 (seasonal average) relative to 1979 -1989 (seasonal average) in (a) Winter (DJF) and (b) Summer (JJA). Source: Era Interim Reanalysis.

**Fig. 1.**

[Figure]

---

## Author Comment (AC3) · 10 Feb 2020

Thank you very much for your consideration. We really appreciate the comments and have learned a lot. In order to improve the analyses as you suggested and also following the indication from referee #1, we add new results: 1) analysis of polar amplification from observational data (Figure 1) and sea ice analysis from different CMIP5 models (Figure 3, Figure 4 and Table 1). This analysis provided greater robustness in the results, which were included here in several parts of the revised manuscript. Thus replacing, expressions as "we suggest" with more complete discussions. Also, appropriate changes were made in the revised manuscript (expanding discussion) according

to the suggestions. 1) Regarding the Climate models chosen, we chose models from different locations considered state-of-the-art climate models (North America - USA (2), Europe – German and France(2), Japan (1), Asutralia (1)). Harrison et al. (2015), in a Nature Climate Change publciation used seven state–of-the-art CMIP5 climate model to explain the evolution of CMIP5 papaeo-simulation to improve climate simulations. In our work, we used pratically the same models, adding – BESM-OA V2.5. Furthermore, GFDL-ESM2M was chosen because has a diferente atmospheric compent, but the same ocean component. 2) Regarding to use CMIP6, we've been working hard to finish our experiments, unfortunately, we are not done yet. 3) Other BESM-OA and state-of-the-art CMIP5 numerical experiments (as RCP and decennial) have been previously published in Casagrande et al. (2016) and Casagrande (2016). 4) We totally agree and appreciate the valuable suggestions. So we added new analyzes and figures. 5) We have improved both, introduction and conclusions on revised manuscript.

Figure 1. Polar Amplification using Long-term observations of Surface Air Temperatures (oC) at 2008-2018 (seasonal average) relative to 1979 -1989 (seasonal average) in (a) Winter (DJF) and (b) Summer (JJA). Source: Era Interim Reanalysis.

Figure 1 shows the enhanced surface warming at high latitudes compared to the rest of globe, with a slightly greater rate of warming in the 20th century. The observed Polar Amplification is not symmetric, most evidence is from Arctic region (during the boreal winter). According to Stocker et al., (2013), the enhanced warming at northern high latitudes was linked with decrease in snow cover and sea ice concentration, sea level rise and increase in land precipitation. Besides that, changes in atmospheric and ocean circulation (Chylek et al., 2019; Pedersen et al., 2016; Pithan and Mauritsen, 2014; Stocker et al., 2013; Yang et al., 2010; Graversen et al., 2008).

Following the reviewer's suggestion and in order to better discuss the relationship between enhanced warming at high latitudes (Figure 1) and sea ice changes, we include the Figure 3, Figure 4 and Table1.

Figure 3 (new - attached here) shows, under the largest future GHG (4xCO2), the spatial pattern of sea ice changes for both, Arctic and Antarctic (difference between sea ice concentration for the last 30 years of abrupt4xCO2 numerical experiment and the last 30 years of the piControl run). This new Figure complements and makes the discussion shown in Figure 1 (old manuscript) more robust. The maximum of the Arctic warming obtained from observations (new Figure 1) and different CMIP5 simulations (old Figure 1) occurs in boreal winter (DJF). According to Figure 1 (old manuscript), the following models, in descending order, appears as having greater amplification: MIROC – ESM, MPI-ESM, BESM-OA V2.5 and CSIRO-ACCESS. Similar response, for the same period is observed in Figure 3 and Figure 4, related to sea ice changes. The large decrease in sea ice concentration is more evident in models with great Polar Amplification, and for the same range of latitude (75o N – 90o N). The end of melting period (when sea ice reaches its minimum annual value) for all models shows sea ice-free conditions. Models that have strong Polar Amplification exhibit expressive changes in the sea ice annual amplitude with outstanding ice-free condition from may to December (MIROC-ESM) and June to December (MPI-ESM). Then, the end of melting period is expected early, likely, associated a large decrease in sea ice thickness and contributing to a delay in sea ice formation. We suggest, based in Figure 3 and Table 1, that, the Arctic will become covered only by first year sea ice (more vulnerable to melting), making the region more sensitive thermodynamically and dynamically to temperature changes. These new evidences presented here, corroborates with the theory, that the Polar Amplification is closely linked to sea ice albedo feedback. For Antarctica, however, the same physical processes cannot be used to explain the Polar Amplification (as discussed in the manuscript). Although, according to Figure 1 (old manuscript) and Figure 3 (new - attached here), there is a small indication of the contribution of sea ice albedo feedback in Antarctic Polar Amplification. Latitudes between 60oN and 65oN (greater Polar Amplification, models BESM-OAV2.5, MIROC-ESM and NCAR-CCSM4) for Austral winter also have trace of relation with abrupt changes in sea ice (Figure 3). Here, it is important to consider the contribution of the ice sheet in Polar Amplification

that is not represented by the most of CMIP5 current models. According to Salzmann (2017 the overall weaker warming in Antarctica is due to a more efficient ocean heat uptake in the southern ocean, weaker surface albedo feedback in combination with ozone depletion.

Figure 3. Sea ice concentration for the last 30 years of Abrupt4xCO2 numerical experiment minus the last 30 years of the piControl run for the following models: BESM-OA V2.5, NCAR-CCSM4, GFDL-ESM-LR, MPI-ESM-LR, CSIRO, IPSL and MIROC-ESM in March (left column) and September (right column).

Table 1. Sea ice area (million square kilometers) for the last 30 years of the abrupt 4xCO2 numerical experiment minus the last 30 years of the piControl run for the following models: BESM-OA V2.5, NCAR-CCSM4, GFDL-ESM-LR, MPI-ESM-LR, CSIRO, IPSL and MIROC-ESM. I Arctic (Antarctic) sea ice reach its annual maximum area in march (february) and the minimum period in September.

Figure 4. Climatology of maximum and minimum Sea ice area (million square kilometers) for the last 30 years of the abrupt 4xCO2 numerical experiment minus the last 30 years of the piControl run for the following models: BESM-OA V2.5, NCAR-CCSM4, GFDL-ESM-LR, MPI-ESM-LR, CSIRO, IPSL and MIROC-ESM. (a) Arctic, (b) Antarctic. Black color represents the maximum (minimum) period of sea ice concentration, march (february) month for Arctic (Antarctic). Gray color bar represents September month.

Specific comments:

Pg. 1; L. 8: "The numerical climate simulation from Brazilian Earth System Model (BESM) are..." – Replace "are" by "is" or "simulation" by "simulations". Reply: ok

Pg. 1; Ls. 18, 19, 21, 24: Consider to add an article in the following cases – "warming at the surface", "heat in the atmosphere.", "for the cold season", and "in the coming decades". Also, for other instances in the manuscript. Reply: ok

Overall comment: For uncountable nouns, the use of the indefinite article "a" may be redundant. For instance: "a warming", "a cooling". This rule could be considered for the entire manuscript. Reply: ok

Pg. 2; Ls. 31: I guess the authors meant GHG rather "GHC Reply: ok

Pg. 2; Ls. 35–39: The sentence is confusing. It is kind of hard to get what the authors mean. Please, consider to rewrite it. For instance, "these two-poles inter-hemispheric asymmetries in the mean ocean circulation" but nothing was mentioned for the "Arctic mean circulation" Reply: ok

Pg. 2; L. 37: "According Marshall..." replace by "According to Marshall". Please, check for the other instances in the text. Reply: ok

Pg. 2; Ls. 40–42: "Numerous..." but only Vaughan was cited. Numerous scientific publications based on both, observations and state-of-the-art Global Climate Model simulations for the high latitudes of the northern hemisphere have shown that AA is an intrinsic feature of the Earth's climate system (Smith et al., 2019; Vaughan et al., 2013; Serreze and Barry, 2011; Screen and Simmonds, 2010).

Pg. 2; Ls. 45–46: "from between 1875 and 2008" – Drop "from". Reply: ok

Pg. 2; Ls. 46–47: Add "the" in "latitudes of the northern hemisphere". Reply: ok

Pg. 2; L. 55: Replace "this processes" by "these processes"; Also, it seems that the explanation "Ocean is becoming more like the Atlantic ocean" is not required. Reply: ok

Pg. 2; L. 59: "The large differences among the models is" – Replace "is" by "are". Reply: ok

Pg. 3; Ls. 78–81: I was wondering why comparing the BESM results against only 5 other models rather than the entire ensemble of models? Also, since we are already in the CMIP6, why not make this study with experiments from this phase. In addition,

since the 4xCO2 seems a bit unrealistic, I think the use of the simulations forced by "1% per year CO2 increase (1pctCO2; Eyring et al., 2016)" would strength the manuscript.

Pg. 3; L. 81: "The paper was is organized". Reply: ok

Pg. 3; L. 86: Missing "." at the end of the sentence. Reply: ok

Pg. 3; L. 93: "an a instantaneous"; "the 21st". There is a mistake with numbering sections as per Sec. 3. Reply: ok Reply: ok

Pg. 5; L. 129: "accesses". Do you mean "assess"? Reply: ok

Pg. 5; L. 128–129: It does not seem to be the case since the discussion for Arctic and Antarctic is, in some instances, merged in Sec. 3 Reply: ok

Pg. 5; L. 135: Replace "assesses" by "assess". Reply: ok

Pg. 6; L. 138: Replace "This procedure been largely" by "This procedure has been". Also, the authors argued "largely" but cited only 2 references. Reply: ok

This procedure has been largely used by researchers since allows us to evaluate and compare potential warming and sensitivities between low and high latitudes as well as to compare differences between models (Van der Linden et al., 2019; Cvijanovic et al., 2015; Manabe et al., 2004; Holand and Bitz, 2003).

Pg. 6; L. 138: "Contrasting, the tropical warming for both, northern and southern hemisphere, is pretty similar with not so accentuated SAT increase in summer and for regions close to 30N." – Not sure I agree with this statement. From Fig. 1, it is noticeable an increase in the SAT differences from about -60S to +60N. Could the authors add some words/explanation for that in the manuscript? Reply: ok

Pg. 6; L. 146–147: ". . . the overall weaker warming in Antarctica is due to a more efficient ocean heat uptake in the southern ocean". I am wondering whether the authors could test this by looking at the SST data (or another output variable). For instance, is the Polar Amplification and respective seasonal cycle also observed in the SST data.

If so, what are the differences between Antarctic and Arctic? Maybe something could be shown in terms of albedo feedback. I think this is a better way to address the issue rather than "We expect...". Reply: ok

Pg. 6; L. 155: "reaching a minimum at 70S" – I would rather say 60S. Reply: ok

Pg. 6; L. 160: "The main reason for winter (DJF) Arctic Amplification pointed by Ser-reze et al., (2009) is largely driven by changes in sea ice, allowing for intense heat transfers from the ocean to the atmosphere...". I also think the authors could check this hypothesis with the CMIP datasets. Reply: ok

Pg. 6; L. 163: Replace "looses" by "loses". Reply: ok

Pg. 7; L. 171: Replace "consequent" by "consequently". Reply: ok

Pg. 7; L. 174–178: The referred teleconnection seems to be out of context here. Reply: ok

Pg. 7; L. 180: Replace "trend" by "tends"(?) Reply: ok

Pg. 7; L. 190: Replace "In the other hand" by "On the other hand" Reply: ok

Pg. 7; L. 197: Replace "Artic" by "Arctic". Reply: ok

Pg. 7; L. 203: Replace "register" by "registered". Reply: ok

Pg. 8; L. 209: Replace "previously version" by "previous version". Reply: ok

Pg. 8; L. 208–212: Not sure the comparison between the two BESM versions makes sense in the scope of the manuscript. The paper compares different models but not different versions of the same model. As it is, it seems like an artifact for auto-citation. Reply: ok

Fig. 2 – I think this analysis should be performed for the ensemble of models. Fig. 3: This figure should be further improved. The labels are too small; it is missing the y-label and unity; the colorbar is not aligned with the figures. Reply: ok

Pg. 11; L. 275: Replace "a combination changes in winds" by "a combination of changes in winds" Reply: ok
* * *
Figure 1. Polar Amplification using Long-term observations of Surface Air Temperatures ($^{o}$C) at 2008-2018 (seasonal average) relative to 1979 -1989 (seasonal average) in (a) Winter (DJF) and (b) Summer (JJA). Source: Era Interim Reanalysis.

[Figure]

**Fig. 1.**

[Figure]

**Fig. 2.**

MAR MIROC-ESM     SEP MIROC-ESM

MAR MPI - ESM     SEP MPI - ESM

MAR NCAR - CCSM4     SEP NCAR - CCSM4

Figure 3. Sea ice concentration for the last 30 years of abrupt4xCO$_2$ numerical experiment minus the last 30 years of the piControl run for the following models: BESM-OA V2.5, NCAR-CCSM4, GFDL-ESM-LR, MPI-ESM-LR, CSIRO-ACCESS, IPSL and MIROC-ESM in March (left column) and September (right column).

**Fig. 3.**

[Figure]

Figure 4. Climatology of maximum and minimum Sea ice area (million square kilometers) for the last 30 years of the abrupt 4x$CO_2$ numerical experiment minus the last 30 years of the piControl run for the following models: BESM-OA V2.5, NCAR-CCSM4, GFDL-ESM-LR, MPI-ESM-LR, CSIRO, IPSL and MIROC-ESM. (a) Arctic, (b) Antarctic. Black color represents the maximum (minimum) period of sea ice concentration, march (february) month for Arctic (Antarctic). Gray color bar represents September month.

**Fig. 4.**

| CMIP5 Models | Arctic | | | Antarctic | | |
|---|---|---|---|---|---|---|
| | | March | Sept | | Feb | Sept |
| BESM-OA | piControl | 16 | 3.5 | piControl | 1 | 29 |
| | 4xCO$_2$ | 11.5 | Ice-Free [Aug-Nov] | 4xCO$_2$ | Ice-Free | 17 |
| | | March | Sept | | Feb | Sept |
| CSIRO ACCESS | piControl | 14 | 6 | piControl | 4.5 | 17 |
| | 4xCO$_2$ | 7 | Ice-Free [Jul-Nov] | 4xCO$_2$ | 1 | 11 |
| | Arctic | March | Sept | Antarctic | Feb | Sept |
| GFDL –ESM2M | piControl | 14 | 7 | piControl | Ice-Free | 9 |
| | 4xCO$_2$ | 9 | Ice-Free | 4xCO$_2$ | Ice-Free [Feb-Mar] | 4 |
| | | March | Sept | | Feb | Sept |
| IPSL –CM5-LR | piControl | 13 | 6 | piControl | 1 | 17 |
| | 4xCO$_2$ | 8.5 | Ice-Free [Jul-Oct] | 4xCO$_2$ | Ice -Free [Jan-Mar] | 7 |
| | | March | Sept | | Feb | Sept |
| MIROC-ESM | piControl | 13 | 6 | piControl | 1 | 14 |
| | 4xCO$_2$ | 0.8 | Ice-Free [May-Dec] | 4xCO$_2$ | Ice free | 8.5 |
| | | March | Sept | | Feb | Sep |
| MPI –ESM | piControl | 12 | 7 | piControl | 1 | 13 |
| | 4xCO$_2$ | 3.5 | Ice-Free [Jun-Dec] | 4xCO$_2$ | Ice-Free [Jan-Apr] | 4 |
| | | March | Sept | | Feb | Mar |
| NCAR –CCSM4 | piControl | 13 | 8 | piControl | 7.5 | 22 |
| | 4xCO$_2$ | 10 | Ice-Free [Aug-Oct] | 4xCO$_2$ | 1 | 11 |

Table 1. Climatology of maximum and minimum Sea ice area (million square kilometers) for the last 30 years of the abrupt 4xCO$_2$ numerical experiment and the last 30 years of the piControl run for the following models: BESM-OA V2.5, NCAR-CCSM4, GFDL-ESM-LR, MPI-ESM-LR, CSIRO, IPSL and MIROC-ESM.

**Fig. 5.**

---

## Author Comment (AC4) · 10 Feb 2020

Casagrande et al. (2019)

Referee #3

Answer:

Thank you very much for your consideration. We really appreciate the comments and have learned a lot. Appropriate changes were made in the revised manuscript according to the suggestions.

In order to improve the analyses, following your suggestion and from referee #1, we

add new results: 1) analysis of polar amplification from observational data (Figure 1) and sea ice analysis from different CMIP5 models (Figure 3, Figure 4 and Table 1). This analysis provided greater robustness in the results, which were included here in several parts of the revised manuscript. Thus replacing, expressions as "we suggest" with more complete discussions. Also, appropriate changes were made in the revised manuscript (expanding discussion) according to the suggestions.

Figure 1. Polar Amplification using Long-term observations of Surface Air Temperatures (oC) at 2008-2018 (seasonal average) relative to 1979 -1989 (seasonal average) in (a) Winter (DJF) and (b) Summer (JJA). Source: Era Interim Reanalysis.

Figure 1 shows the enhanced surface warming at high latitudes compared to the rest of globe, with a slightly greater rate of warming in the 20th century. The observed Polar Amplification is not symmetric, most evidence is from Arctic region (during the boreal winter). According to Stocker et al., (2013), the enhanced warming at northern high latitudes was linked with decrease in snow cover and sea ice concentration, sea level rise and increase in land precipitation. Besides that, changes in atmospheric and ocean circulation (Chylek et al., 2019; Pedersen et al., 2016; Pithan and Mauritsen, 2014; Stocker et al., 2013; Yang et al., 2010; Graversen et al., 2008).

Following the reviewer's suggestion and in order to better discuss the relationship between enhanced warming at high latitudes (Figure 1) and sea ice changes, we include the Figure 3, Figure 4 and Table1.

Figure 3 (new - attached here) shows, under the largest future GHG (4xCO2), the spatial pattern of sea ice changes for both, Arctic and Antarctic (difference between sea ice concentration for the last 30 years of abrupt4xCO2 numerical experiment and the last 30 years of the piControl run). This new Figure complements and makes the discussion shown in Figure 1 (old manuscript) more robust. The maximum of the Arctic warming obtained from observations (new Figure 1) and different CMIP5 simulations (old Figure 1) occurs in boreal winter (DJF). According to Figure 1 (old manuscript),

the following models, in descending order, appears as having greater amplification: MIROC – ESM, MPI-ESM, BESM-OA V2.5 and CSIRO-ACCESS. Similar response, for the same period is observed in Figure 3 and Figure 4, related to sea ice changes. The large decrease in sea ice concentration is more evident in models with great Polar Amplification, and for the same range of latitude (75o N – 90o N). The end of melting period (when sea ice reaches its minimum annual value) for all models shows sea ice-free conditions. Models that have strong Polar Amplification exhibit expressive changes in the sea ice annual amplitude with outstanding ice-free condition from may to December (MIROC-ESM) and June to December (MPI-ESM). Then, the end of melting period is expected early, likely, associated a large decrease in sea ice thickness and contributing to a delay in sea ice formation. We suggest, based in Figure 3 and Table 1, that, the Arctic will become covered only by first year sea ice (more vulnerable to melting), making the region more sensitive thermodynamically and dynamically to temperature changes. These new evidences presented here, corroborates with the theory, that the Polar Amplification is closely linked to sea ice albedo feedback. For Antarctica, however, the same physical processes cannot be used to explain the Polar Amplification (as discussed in the manuscript). Although, according to Figure 1 (old manuscript) and Figure 3 (new - attached here), there is a small indication of the contribution of sea ice albedo feedback in Antarctic Polar Amplification. Latitudes between 60oN and 65oN (greater Polar Amplification, models BESM-OAV2.5, MIROC-ESM and NCAR-CCSM4) for Austral winter also have trace of relation with abrupt changes in sea ice (Figure 3). Here, it is important to consider the contribution of the ice sheet in Polar Amplification that is not represented by the most of CMIP5 current models. According to Salzmann (2017 the overall weaker warming in Antarctica is due to a more efficient ocean heat uptake in the southern ocean, weaker surface albedo feedback in combination with ozone depletion.

Figure 3. Sea ice concentration for the last 30 years of Abrupt4xCO2 numerical experiment minus the last 30 years of the piControl run for the following models: BESM-OA V2.5, NCAR-CCSM4, GFDL-ESM-LR, MPI-ESM-LR, CSIRO, IPSL and MIROC-ESM

in March (left column) and September (right column).

Table 1. Sea ice area (million square kilometers) for the last 30 years of the abrupt 4xCO2 numerical experiment minus the last 30 years of the piControl run for the following models: BESM-OA V2.5, NCAR-CCSM4, GFDL-ESM-LR, MPI-ESM-LR, CSIRO, IPSL and MIROC-ESM. I Arctic (Antarctic) sea ice reach its annual maximum area in march (february) and the minimum period in September.

Figure 4. Climatology of maximum and minimum Sea ice area (million square kilometers) for the last 30 years of the abrupt 4xCO2 numerical experiment minus the last 30 years of the piControl run for the following models: BESM-OA V2.5, NCAR-CCSM4, GFDL-ESM-LR, MPI-ESM-LR, CSIRO, IPSL and MIROC-ESM. (a) Arctic, (b) Antarctic. Black color represents the maximum (minimum) period of sea ice concentration, march (february) month for Arctic (Antarctic). Gray color bar represents September month.

1) In relation to the objective, we changed to: The main objective is to investigate the seasonality of the surface and vertical warming, the seasonal response of sea ice, as well as the coupled processes underlying the polar amplification.

2) We have improved both, introduction and conclusions on revised manuscript including the new results and as suggested by the referee.

Specific Comments: Page 2, L. 40: "Numerous Scientific Publications"? I suggest rewriting this paragraph because it is confusing. Reply: ok

Numerous scientific publications based on both, observations and state-of-the-art Global Climate Model simulations for the high latitudes of the northern hemisphere have shown that AA is an intrinsic feature of the Earth's climate system (Smith et al., 2019; Vaughan et al., 2013; Serreze and Barry, 2011; Screen and Simmonds, 2010).

Page 2, L. 56: References ..? Reply: ok

Page 5, L. 129: Replaced "parsed" with "parsed" Reply: ok

Page, L. 132: Attention to section description: 3.1 Polar ... Reply: ok
* * *
[Figure]

Winter DJF Obs (2008−2018 minos 1979−1989)

SummerJJA Obs (2008−2018 minos 1979−1989)

Figure 1. Polar Amplification using Long-term observations of Surface Air Temperatures ($^{o}$C) at 2008-2018 (seasonal average) relative to 1979 -1989 (seasonal average) in (a) Winter (DJF) and (b) Summer (JJA). Source: Era Interim Reanalysis.

**Fig. 1.**

[Figure]

**Fig. 2.**

[Figure]

Figure 3. Sea ice concentration for the last 30 years of abrupt4xCO$_2$ numerical experiment minus the last 30 years of the piControl run for the following models: BESM-OA V2.5, NCAR-CCSM4, GFDL-ESM-LR, MPI-ESM-LR, CSIRO-ACCESS, IPSL and MIROC-ESM in March (left column) and September (right column).

**Fig. 3.**

[Figure]

Figure 4. Climatology of maximum and minimum Sea ice area (million square kilometers) for the last 30 years of the abrupt $4xCO_2$ numerical experiment minus the last 30 years of the piControl run for the following models: BESM-OA V2.5, NCAR-CCSM4, GFDL-ESM-LR, MPI-ESM-LR, CSIRO, IPSL and MIOC-ESM. (a) Arctic, (b) Antarctic. Black color represents the maximum (minimum) period of sea ice concentration, march (february) month for Arctic (Antarctic). Gray color bar represents September month.

**Fig. 4.**

| CMIP5 Models | Arctic | | | Antarctic | | |
|---|---|---|---|---|---|---|
| **BESM-OA** | | March | Sept | | Feb | Sept |
| | **piControl** | 16 | 3.5 | **piControl** | 1 | 29 |
| | **4xCO₂** | 11.5 | Ice-Free [Aug-Nov] | **4xCO₂** | Ice-Free | 17 |
| **CSIRO ACCESS** | | March | Sept | | Feb | Sept |
| | **piControl** | 14 | 6 | **piControl** | 4.5 | 17 |
| | **4xCO₂** | 7 | Ice-Free [Jul-Nov] | **4xCO₂** | 1 | 11 |
| **GFDL –ESM2M** | Arctic | March | Sept | Antarctic | Feb | Sept |
| | **piControl** | 14 | 7 | **piControl** | Ice-Free | 9 |
| | **4xCO₂** | 9 | Ice-Free | **4xCO₂** | Ice-Free [Feb-Mar] | 4 |
| **IPSL –CM5-LR** | | March | Sept | | Feb | Sept |
| | **piControl** | 13 | 6 | **piControl** | 1 | 17 |
| | **4xCO₂** | 8.5 | Ice-Free [Jul-Oct] | **4xCO₂** | Ice -Free [Jan-Mar] | 7 |
| **MIROC-ESM** | | March | Sept | | Feb | Sept |
| | **piControl** | 13 | 6 | **piControl** | 1 | 14 |
| | **4xCO₂** | 0.8 | Ice-Free [May-Dec] | **4xCO₂** | Ice free | 8.5 |
| **MPI –ESM** | | March | Sept | | Feb | Sep |
| | **piControl** | 12 | 7 | **piControl** | 1 | 13 |
| | **4xCO₂** | 3.5 | Ice-Free [Jun-Dec] | **4xCO₂** | Ice-Free [Jan-Apr] | 4 |
| **NCAR –CCSM4** | | March | Sept | | Feb | Mar |
| | **piControl** | 13 | 8 | **piControl** | 7.5 | 22 |
| | **4xCO₂** | 10 | Ice-Free [Aug-Oct] | **4xCO₂** | 1 | 11 |

Table 1. Climatology of maximum and minimum Sea ice area (million square kilometers) for the last 30 years of the abrupt $4xCO_2$ numerical experiment and the last 30 years of the piControl run for the following models: BESM-OA V2.5, NCAR-CCSM4, GFDL-ESM-LR, MPI-ESM-LR, CSIRO, IPSL and MIROC-ESM.

**Fig. 5.**

---

## Author Comment (AC5) · 10 Feb 2020

In relation to: Fig. 3: This figure should be further improved. The labels are too small; it is missing the y-label and unity; the colorbar is not aligned with the figures.

[Figure]

Figure 5. Zonal-average atmosphere temperature changes, in °C (Abrupt 4xCO$_2$ minus piControl) at each pressure level, in mb (solid line) for the last 30 years run for (a) BESM OA V2.5, (b) NCAR-CCSM4 and (c) MPI-ESM-LR model, in DJF (left) and JJA (right) columns.

**Fig. 1.**

---

## Author Response (AR1)

**Casagrande et al. (2019).**

**Referee #1**

**Comments to Authors**

The authors analyzed the seasonality of polar amplification using CMIP5 coupled climate models and BESM to compare 4xCO2 and piControl experiments. The results show that there is an asymmetry in the polar amplifications between the two hemispheres. I do believe that the scientific subject and aims of this paper are within the scope of ANGEO and this paper is easy to read, but I also think that this specific paper could need a significant amount of new analysis and modeling work in order to appropriately address its hypotheses.

**Major comments:**

I am not sure whether this paper is a letter or an article. If it is a letter paper, I didn't see relatively interesting points worthy of publication on ANGEO in the present version. If it is an article paper, I think it should include more analysis results. For example, it will be better to show the asymmetry in the seasonality of polar amplification in different hemispheres in observations. The authors cited many references and used a lot of 'suggest' to explain the figures. However, I think it may be better to give a direct evidence and make more figures to support them, e.g., L197 sea ice patterns and L237 plotting atmospheric heat transport in different models, etc.

**Answer:**

Thank you very much for your consideration. We really appreciate the comments and have learned a lot. Appropriate changes were made in the revised manuscript according to the suggestions.

Here, we include new Figures and two attachments files:
File 1: old document with changes indicated (in red): Word_track_changes_casagrande_file2.pdf
File 2: revised document: Casagrande_file1_word_final.pdf

First, this paper is an article. We added results as you suggest, for example: analysis of polar amplification from observational data (Figure 1) and sea ice analysis from different CMIP5 models (Figure 4 and Table 1). This analysis provided greater robustness in the results, which were included here in several parts of the revised manuscript. Thus replacing, expressions as "we suggest" with more complete discussions.

Figure 1. Polar Amplification using Long-term observations of Surface Air Temperatures ($^{o}$C) at 2008-2018 (seasonal average) relative to 1979 -1989 (seasonal average) in (a) Winter (DJF) and (b) Summer (JJA). Source: Era Interim Reanalysis.

Figure 1 shows the enhanced surface warming at high latitudes compared to the rest of globe, with a slightly greater rate of warming in the 20th century. The observed Polar Amplification is not symmetric, most evidence is from Arctic region (during the boreal winter). According to Stocker et al., (2013), the enhanced warming at northern high latitudes was linked with decrease in snow cover and sea ice concentration, sea level rise and increase in land precipitation. Besides that, changes in atmospheric and ocean circulation (Chylek et al., 2019; Pedersen et al., 2016; Pithan and Mauritsen, 2014; Stocker et al., 2013; Yang et al., 2010; Graversen et al., 2008). For paleo-climate periods, the

Polar Amplification also is reported by Climate Models, driven by solar or natural carbon cycle perturbations (Sundqvist et al., 2010; O'ishi and Abe-Ouchi, 2011; Mann et al., 2009; Masson-Delmotte et al, 2006)

Following the reviewer's suggestion and in order to better discuss the relationship between enhanced warming at high latitudes (Figure 1) and sea ice changes, we include the Figure 4, and Table 1.

Figure 4 (new - attached here) shows, under the largest future GHG ($4xCO_2$), the spatial pattern of sea ice changes for both, Arctic and Antarctic (difference between sea ice concentration for the last 30 years of abrupt$4xCO_2$ numerical experiment and the last 30 years of the piControl run). This new Figure complements and makes the discussion shown in Figure 1 (old manuscript) more robust. The maximum of the Arctic warming obtained from observations (new Figure 1) and different CMIP5 simulations (old Figure 1) occurs in boreal winter (DJF). According to Figure 1 (old manuscript), the following models, in descending order, appears as having greater amplification: MIROC – ESM, MPI-ESM, BESM-OA V2.5 and CSIRO-ACCESS. Similar response, for the same period is observed in Figure 3 and Figure 4, related to sea ice changes. The large decrease in sea ice concentration is more evident in models with great Polar Amplification, and for the same range of latitude ($75^o$ N – $90^o$ N). The end of melting period (when sea ice reaches its minimum annual value) for all models shows sea ice-free conditions. Models that have strong Polar Amplification exhibit expressive changes in the sea ice annual amplitude with outstanding ice-free condition from may to December (MIROC-ESM) and June to December (MPI-ESM). Then, the end of melting period is expected early, likely, associated a large decrease in sea ice thickness and contributing to a delay in sea ice formation. We suggest, based in Figure 4 and Table 1, that, the Arctic will become covered only by first year sea ice (more vulnerable to melting), making the region more sensitive thermodynamically and dynamically to temperature changes. These new evidences presented here, corroborates with the theory, that the Polar Amplification is closely linked to sea ice albedo feedback. For Antarctica, however, the same physical processes cannot be used to explain the Polar Amplification (as discussed in the manuscript). Although, according to Figure 1 (old manuscript) and Figure 4 (new - attached here), there is a small indication of the contribution of sea ice albedo feedback in Antarctic Polar Amplification. Latitudes between $60^o$ N and $65^o$ N (greater Polar Amplification, models BESM-OAV2.5, MIROC-ESM and NCAR-CCSM4) for Austral winter also have trace of relation with abrupt changes in sea ice (Figure 4). Here, it is important to consider the contribution of the ice sheet in Polar Amplification that is not represented by the most of CMIP5 current models. According to Salzmann (2017 the overall weaker warming in Antarctica is due to a more efficient ocean heat uptake in the southern ocean, weaker surface albedo feedback in combination with ozone depletion.

Figure 4. Sea ice concentration for the last 30 years of Abrupt$4xCO_2$ numerical experiment minus the last 30 years of the piControl run for the following models: BESM-OA V2.5, NCAR-CCSM4, GFDL-ESM-LR, MPI-ESM-LR, CSIRO, IPSL and MIROC-ESM in March (left column) and September (right column).

Table 1. Sea ice area (million square kilometers) for the last 30 years of the abrupt $4xCO_2$ numerical experiment minus the last 30 years of the piControl run for the following models: BESM-OA V2.5, NCAR-CCSM4, GFDL-ESM-LR, MPI-ESM-LR, CSIRO, IPSL and MIROC-ESM. I Arctic (Antarctic) sea ice reach its annual maximum area in march (february) and the minimum period in September.

Figure 5. Climatology of maximum and minimum Sea ice area (million square kilometers) for the last 30 years of the abrupt 4xCO2 numerical experiment minus the last 30 years of the piControl run for

the following models: BESM-OA V2.5, NCAR-CCSM4, GFDL-ESM-LR, MPI-ESM-LR, CSIRO, IPSL and MIROC-ESM. (a) Arctic, (b) Antarctic. Black color represents the maximum (minimum) period of sea ice concentration, march (february) month for Arctic (Antarctic). Gray color bar represents September month.

**Specific comments:**
 L40: You mentioned 'numerous scientific publications', but there is only one reference in the end of this sentence.
Reply: ok   Line 70.
Numerous scientific publications based on both, observations and state-of-the-art Global Climate Model simulations for the high latitudes of the northern hemisphere have shown that AA is an intrinsic feature of the Earth's climate system (Smith et al., 2019; Vaughan et al., 2013; Serreze and Barry, 2011; Screen and Simmonds, 2010).

L42: suggest->suggested
Reply: ok  Line 72.

L63: Please give explanations regarding why the performance of Arctic simulation is better.
Reply: ok   L 95.
        According to Shu et al., (2015), Global Climate Models simulations in general offer much better simulations for the Arctic than for the Antarctica. Turner et al., (2015) suggested that the main p roblem of climate models in the high latitudes of the southern hemisphere is their inability to reproduce the observed (although slight) increase in Sea Ice Extent (SIE). Bintanja et al., (2015) and Swart and Fyfe, (2013) have demonstrated the importance to include the effect of the increasing freshwater input from Antarctic continental ice into the Southern Ocean. The authors describe that the ice sheet dynamics, essential for having accurate sea ice simulations, is currently disregarded in all CMIP5 models. Swart and Fyfe (2013) also suggested that this deficiency may significantly influence the simulated sea ice trend because the subsurface ocean warming causes basal ice-shelf melt, freshening the surface waters, which eventually leads to an increase in sea ice formation.. Moreover, the instrumental network for data collection in Antarctica and the Southern Ocean is considered scarce (even more than in the Arctic), inhomogeneous and insufficiently dense to validate climate models. Therefore, or the high latitudes regions of the southern hemisphere, the effects of the ongoing climate change and its associated processes are still considered hot topics that lack conclusive answers.

L75: are also depended on-> depend on
Reply: ok   Line 120.

L76: making -> which makes
Reply: ok  Line 124.

L105: last -> latest
Reply: ok  Line 193.

L158: 'with no so enhanced warming' is confusing. Please rephrase it.
Reply: ok   Line 276.

From March to August, the reverse signal shows the maximum warming close to 70°S, decreasing towards to tropical region, lacking the enhanced warming at the northern high latitudes.

L163: looses heat to -> heats
Reply: ok  Line 294.

L174-L178: Why the authors mentioned the linkage between Arctic sea ice loss and mid-latitude weather? I think it is irrelevant to your topic.
Reply: ok , it is out.

L197: It may be better to have a spatial distribution of sea ice trends to support your hypothesis, see my major comments.
Reply: ok , Figure 4, Figure 5 and Table 1.

L209: last->latest
Reply: ok

L247: It is difficult for me to link Fig.3 to deep convection. I think the authors should give more evidences to support your conclusions. See my major comments.

Reply: ok, we agree with suggestion.

**Referee #2**

Thank you for your constructive and helpful feedback. We really appreciate the comments and have learned a lot. Appropriate changes were made in the revised manuscript according to the suggestions.

**Comments to Authors**

The authors studied the seasonality of the Polar Amplification (PA), here defined as the difference between the CMIP5 control (piControl) and the abrupt increased CO2 (abrupt4xCO2) experiments. The manuscript shows that the Arctic is more sensible to the PA, which is more remarkable in autumn and winter. I think this is a timely and interesting topic that certainly deserves attention from the scientific community. Also, the paper fits well in the short-communication format (I am assuming this is the case). In my opinion, the manuscript has potential, but the analyses presented so far are too shallow and it should be further improved before publication. First, I think the authors should use all the CMIP models available, and not only 6 of them: CMIP5 has 31 models that performed the abrupt4xCO2 experiment. If there is a good reason to use only these 6 models, this should be acknowledged in the text. By using a larger number of models, the authors would be able to perform some statistical analysis (e.g., to compare BESM against the others) and bring robustness to the manuscript. Second, it would be much more useful to the scientific community to see this study conducted with the CMIP6 outputs. Again, the data processing and analyses performed so far are straightforward so that it should not be a problem to adapt them to the CMIP6 models. Third, although I understand that is fair to use the abrupt 4xCO2 experiments in this study, the authors could bring other experiments to their analyses such as the 1pctCO2. Fourth, as the first reviewer also pointed out, I also think that many of the hypotheses raised by the authors could be effectively tested with the CMIP data. Fifth, I miss in the introduction a strong point on what this manuscript brings as new results and, I also missed a more comprehensive conclusion for the new findings. Finally, a bit more care with manuscript writing is required. I have pointed out some mistakes below (not exhaustively), as well as other comments that could be considered by the authors for improving their manuscript.

**Answer:**

Thank you very much for your consideration. We really appreciate the comments and have learned a lot.

Here, we include news Figures and two attachments files:
**File 1:** old document with changes indicated (in red): Word_track_changes_casagrande_file2.pdf
**File 2:** revised document: Casagrande_file1_word_final.pdf

In order to improve the analyses and following the suggestion from referee #1, we add new results: 1) analysis of polar amplification from observational data (Figure 1) and sea ice analysis from different CMIP5 models (Figure 4, Figure 5 and Table 1). This analysis provided greater robustness in the results, which were included here in several parts of the revised manuscript. Thus replacing, expressions as "we suggest" with more complete discussions. Also, appropriate changes were made in the revised manuscript (expanding discussion) according to the suggestions (Casagrande_file1_word_final.pdf and Word_track_changes_casagrande_file2.pdf).

Regarding the Climate models chosen, we chose models from different locations, considered state-of-the-art climate models (North America - USA (2), Europe – German and France (2), Japan (1), Australia (1)). Harrison et al. (2015), in a Nature Climate Change publication used seven state–of-the-art CMIP5 climate model to explain the evolution of CMIP5 paleo-simulation to improve climate simulations. In our work, we used practically the same models, adding – BESM-OA V2.5. Furthermore, GFDL-ESM2M was chosen because has a diferente atmospheric compent, but the same ocean component. Regarding to use CMIP6, we've been working hard to finish our experiments, unfortunately, we are not done yet. Other BESM-OA and other state-of-the-art CMIP5 numerical experiments (as RCP and decennial) have been previously published in Casagrande et al. (2016) and Casagrande (2016). In this work, our goal is analyze the climate sensitivity and how amplified can be the response of the polar region in comparison to the globe as a whole, for this, we choose the abrupt $4xCO_2$ experiment. We totally agree and appreciate the valuable suggestions. So we added new analyzes and Figures. Also, we have improved both, introduction and conclusions on revised manuscript.

Figure 1 shows the enhanced surface warming at high latitudes compared to the rest of globe, with a slightly greater rate of warming in the 20th century. The observed Polar Amplification is not symmetric, most evidence is from Arctic region (during the boreal winter). According to Stocker et al., (2013), the enhanced warming at northern high latitudes was linked with decrease in snow cover and sea ice concentration, sea level rise and increase in land precipitation. Besides that, changes in atmospheric and ocean circulation (Chylek et al., 2019; Pedersen et al., 2016; Pithan and Mauritsen, 2014; Stocker et al., 2013; Yang et al., 2010; Graversen et al., 2008). For paleo-climate periods, the Polar Amplification also is reported by Climate Models, driven by solar or natural carbon cycle perturbations (Sundqvist et al., 2010; O'ishi and Abe-Ouchi, 2011; Mann et al., 2009; Masson-Delmotte et al, 2006)

Following the reviewer's suggestion and in order to better discuss the relationship between enhanced warming at high latitudes (Figure 1) and sea ice changes, we include the Figure 3, Figure 4 and Table1.

Here, we include new Figures and two attachments files: File 1: old document with changes indicated (in red) and File 2: revised document.

Figure 4 (new - attached here) shows, under the largest future GHG ($4xCO_2$), the spatial pattern of sea ice changes for both, Arctic and Antarctic (difference between sea ice concentration for the last 30 years of abrupt4xCO2 numerical experiment and the last 30 years of the piControl run). This new Figure complements and makes the discussion shown in Figure 1 (old manuscript) more robust. The maximum of the Arctic warming obtained from observations (new Figure 1) and different CMIP5 simulations (old Figure 1) occurs in boreal winter (DJF). According to Figure 1 (old manuscript), the following models, in descending order, appears as having greater amplification: MIROC – ESM, MPI-ESM, BESM-OA V2.5 and CSIRO-ACCESS. Similar response, for the same period is observed in Figure 3 and Figure 4, related to sea ice changes. The large decrease in sea ice concentration is more evident in models with great Polar Amplification, and for the same range of latitude ($75^o N – 90^o N$). The end of melting period (when sea ice reaches its minimum annual value) for all models shows sea ice-free conditions. Models that have strong Polar Amplification exhibit expressive changes in the sea ice annual amplitude with outstanding ice-free condition from may to December (MIROC-ESM) and June to December (MPI-ESM). Then, the end of melting period is expected early, likely, associated a large decrease in sea ice thickness and contributing to a delay in sea ice formation. We suggest, based in Figure 4 and Table 1, that, the Arctic will become covered only by first year sea ice (more vulnerable to melting), making the region more sensitive thermodynamically and dynamically to

temperature changes. These new evidences presented here, corroborates with the theory, that the Polar Amplification is closely linked to sea ice albedo feedback. For Antarctica, however, the same physical processes cannot be used to explain the Polar Amplification (as discussed in the manuscript). Although, according to Figure 1 (old manuscript) and Figure 4 (new - attached here), there is a small indication of the contribution of sea ice albedo feedback in Antarctic Polar Amplification. Latitudes between $60^{o}$N and $65^{o}$N (greater Polar Amplification, models BESM-OAV2.5, MIROC-ESM and NCAR-CCSM4) for Austral winter also have trace of relation with abrupt changes in sea ice (Figure 4). Here, it is important to consider the contribution of the ice sheet in Polar Amplification that is not represented by the most of CMIP5 current models. According to Salzmann (2017 the overall weaker warming in Antarctica is due to a more efficient ocean heat uptake in the southern ocean, weaker surface albedo feedback in combination with ozone depletion.

Figure 1. Polar Amplification using Long-term observations of Surface Air Temperatures ($^{o}$C) at 2008-2018 (seasonal average) relative to 1979 -1989 (seasonal average) in (a) Winter (DJF) and (b) Summer (JJA). Source: Era Interim Reanalysis.

Figure 3. Sea ice concentration for the last 30 years of Abrupt4x$CO_2$ numerical experiment minus the last 30 years of the piControl run for the following models: BESM-OA V2.5, NCAR-CCSM4, GFDL-ESM-LR, MPI-ESM-LR, CSIRO, IPSL and MIROC-ESM in March (left column) and September (right column).

Table 1. Sea ice area (million square kilometers) for the last 30 years of the abrupt 4x$CO_2$ numerical experiment minus the last 30 years of the piControl run for the following models: BESM-OA V2.5, NCAR-CCSM4, GFDL-ESM-LR, MPI-ESM-LR, CSIRO, IPSL and MIROC-ESM. I Arctic (Antarctic) sea ice reach its annual maximum area in march (february) and the minimum period in September.

Figure 4. Climatology of maximum and minimum Sea ice area (million square kilometers) for the last 30 years of the abrupt 4xCO2 numerical experiment minus the last 30 years of the piControl run for the following models: BESM-OA V2.5, NCAR-CCSM4, GFDL-ESM-LR, MPI-ESM-LR, CSIRO, IPSL and MIROC-ESM. (a) Arctic, (b) Antarctic. Black color represents the maximum (minimum) period of sea ice concentration, march (february) month for Arctic (Antarctic). Gray color bar represents September month.

**Specific comments:**

Pg. 1; L. 8: "The numerical climate simulation from Brazilian Earth System Model (BESM) are..." – Replace "are" by "is" or "simulation" by "simulations".
Reply: ok. Line 8.

Pg. 1; Ls. 18, 19, 21, 24: Consider to add an article in the following cases – "warming at the surface", "heat in the atmosphere.", "for the cold season", and "in the coming decades". Also, for other instances in the manuscript.
Reply: ok, we change it.

Overall comment: For uncountable nouns, the use of the indefinite article "a" may be redundant. For instance: "a warming", "a cooling". This rule could be considered for the entire manuscript.
Reply: ok

Pg. 2; Ls. 31: I guess the authors meant GHG rather "GHC
Reply: ok, yes, we change it.

Pg. 2; Ls. 35–39: The sentence is confusing. It is kind of hard to get what the authors mean. Please, consider to rewrite it. For instance, "these two-poles inter-hemispheric asymmetries in the mean ocean circulation" but nothing was mentioned for the "Arctic mean circulation"
Reply: ok.  Line 50-55

Pg. 2; L. 37: "According Marshall..." replace by "According to Marshall". Please, check for the other instances in the text.
Reply: ok.

Pg. 2; Ls. 40–42: "Numerous..." but only Vaughan was cited.
Numerous scientific publications based on both, observations and state-of-the-art Global Climate Model simulations for the high latitudes of the northern hemisphere have shown that AA is an intrinsic feature of the Earth's climate system (Smith et al., 2019; Vaughan et al., 2013; Serreze and Barry, 2011; Screen and Simmonds, 2010).

Pg. 2; Ls. 45–46: "from between 1875 and 2008" – Drop "from".
Reply: ok.  Line 76.

Pg. 2; Ls. 46–47: Add "the" in "latitudes of the northern hemisphere".
Reply: ok.   Line 77.

Pg. 2; L. 55: Replace "this processes" by "these processes"; Also, it seems that the explanation "Ocean is becoming more like the Atlantic ocean" is not required.
Reply: ok Line 85.

Pg. 2; L. 59: "The large differences among the models is" – Replace "is" by "are".
Reply: ok  Line 93.

Pg. 3; Ls. 78–81: I was wondering why comparing the BESM results against only 5 other models rather than the entire ensemble of models? Also, since we are already in the CMIP6, why not make this study with experiments from this phase. In addition, since the 4xCO2 seems a bit unrealistic, I think the use of the simulations forced by "1% per year CO2 increase (1pctCO2; Eyring et al., 2016)" would strength the manuscript.

Pg. 3; L. 81: "The paper was is organized".
Reply: ok. Line 130.

Pg. 3; L. 86: Missing "." at the end of the sentence.
Reply: ok. Line 165

Pg. 3; L. 93: "an a instantaneous"; "the 21st". There is a mistake with numbering sections as per Sec. 3.
Reply: ok. Line 173.
Reply: ok. Line 173.

Pg. 5; L. 129: "accesses". Do you mean "assess"?
Reply: ok.  Line 223.

Pg. 5; L. 128–129: It does not seem to be the case since the discussion for Arctic and Antarctic is, in some instances, merged in Sec. 3
Reply: ok.

Pg. 5; L. 135: Replace "assesses" by "assess".
Reply: ok.

Pg. 6; L. 138: Replace "This procedure been largely" by "This procedure has been". Also, the authors argued "largely" but cited only 2 references.
Reply: ok.  Line 252-256

This procedure has been largely used by researchers since allows us to evaluate and compare potential warming and sensitivities between low and high latitudes as well as to compare differences between models (Van der Linden et al., 2019; Cvijanovic et al., 2015; Manabe et al., 2004; Holand and Bitz, 2003).

Pg. 6; L. 138: "Contrasting, the tropical warming for both, northern and southern hemisphere, is pretty similar with not so accentuated SAT increase in summer and for regions close to 30N." – Not sure I agree with this statement. From Fig. 1, it is noticeable an increase in the SAT differences from about -60S to +60N. Could the authors add some words/explanation for that in the manuscript?
Reply: ok.  Line 261.

Pg. 6; L. 146–147: ". . . the overall weaker warming in Antarctica is due to a more efficient ocean heat uptake in the southern ocean". I am wondering whether the authors could test this by looking at the SST data (or another output variable). For instance, is the Polar Amplification and respective seasonal cycle also observed in the SST data. If so, what are the differences between Antarctic and Arctic? Maybe something could be shown in terms of albedo feedback. I think this is a better way to address the issue rather than "We expect...".
Reply: we include, as referee 1 suggestion, the Figure 4, Figure 5 and Table 1.

 Pg. 6; L. 155: "reaching a minimum at 70S" – I would rather say 60S.
Reply: ok.  Line 273.

Pg. 6; L. 160: "The main reason for winter (DJF) Arctic Amplification pointed by Serreze et al., (2009) is largely driven by changes in sea ice, allowing for intense heat transfers from the ocean to the atmosphere...". I also think the authors could check this hypothesis with the CMIP datasets.
Reply: ok.

Pg. 6; L. 163: Replace "looses" by "loses".
Reply: ok  Line 295.

 Pg. 7; L. 171: Replace "consequent" by "consequently".
Reply: ok.  We change it.

Pg. 7; L. 174–178: The referred teleconnection seems to be out of context here.
Reply: ok

Pg. 7; L. 180: Replace "trend" by "tends"(?)
Reply: ok

Pg. 7; L. 190: Replace "In the other hand" by "On the other hand"
Reply: ok  Line 316.

Pg. 7; L. 197: Replace "Artic" by "Arctic".
Reply: ok  Line 352

Pg. 7; L. 203: Replace "register" by "registered".
Reply: ok  Line 360.

Pg. 8; L. 209: Replace "previously version" by "previous version".
Reply: ok

Pg. 8; L. 208–212: Not sure the comparison between the two BESM versions makes sense in the scope of the manuscript. The paper compares different models but not different versions of the same model. As it is, it seems like an artifact for auto-citation.
Reply: ok,  we change it.

Fig. 2 – I think this analysis should be performed for the ensemble of models. Fig. 3: This figure should be further improved. The labels are too small; it is missing the y-label and unity; the colorbar is not aligned with the figures.
Reply: ok, we change it.

Pg. 11; L. 275: Replace "a combination changes in winds" by "a combination of changes in winds"
Reply: ok  Line 672.

**Referee #3**

Thank you very much for your consideration. We really appreciate the comments and have learned a lot. Appropriate changes were made in the revised manuscript according to the suggestions.

**Comments to Authors**

Major Comments: The authors present here the seasonality of polar amplification (PA) defined as the difference between the different numerical models. I believe the article has a lot of potential because its results show the importance of these analyzes for these regions, and also a greater approach on the subject. However, the article should be enhanced for future publication. For example, the objective is not clear in the Summary. There is no more detailed description of the main objectives that will be addressed in the work. Although the objective is described in the introduction, I find it necessary to present this objective also in the abstract of the article. In addition, the results presented also require a more refined discussion. More details, more comparisons are needed for the new version. The conclusions also need to be improved by showing the importance of the work, a well-explanatory summary of the results ...

**Answer:**

Thank you very much for your consideration. We really appreciate the comments and have learned a lot. Appropriate changes were made in the revised manuscript according to the suggestions. In order to improve the analyses, following your suggestion and from referee #1/2, we add new results: 1) analysis of polar amplification from observational data (Figure 1) and sea ice analysis from different CMIP5 models (Figure 3, Figure 4 and Table 1). This analysis provided greater robustness in the results, which were included here in several parts of the revised manuscript. Thus replacing, expressions as "we suggest" with more complete discussions. Also, appropriate changes were made in the revised manuscript (expanding discussion) according to the suggestions.

Here, we include news Figures and two attachments files:
**File 1:** old document with changes indicated (in red): Word_track_changes_casagrande_file2.pdf
**File 2:** revised document: Casagrande_file1_word_final.pdf

1) In relation to the objective, we changed to:
The main objective is to investigate the seasonality of the surface and vertical warming, the seasonal response of sea ice, as well as the coupled processes underlying the polar amplification.

2) We have improved both, introduction and conclusions on revised manuscript including the new results and as suggested by the referee.

Following the reviewer's suggestion and in order to better discuss the relationship between enhanced warming at high latitudes (Figure 1) and sea ice changes, we include the Figure 3, Figure 4 and Table1.

Here, we include new Figures and two attachments files: File 1: old document with changes indicated (in red) and File 2: revised document.

[revised manuscript text omitted]

**Specific Comments:**
Page 2, L. 40: "Numerous Scientific Publications"? I suggest rewriting this paragraph because it is confusing.
Reply: ok

Numerous scientific publications based on both, observations and state-of-the-art Global Climate Model simulations for the high latitudes of the northern hemisphere have shown that AA is an intrinsic feature of the Earth's climate system (Smith et al., 2019; Vaughan et al., 2013; Serreze and Barry, 2011; Screen and Simmonds, 2010).

Page 2, L. 56: References ..?
Reply: ok

Page 5, L. 129: Replaced "parsed" with "parsed"
Reply: ok

Page, L. 132: Attention to section description: 3.1 Polar ...
Reply: ok, we change it.

---

## Author Response (AR2)

**Response to Referee 1.**

[Figure]

Report #1
Submitted on 30 Apr 2020
Anonymous Referee #1

Anonymous during peer-review: Yes No
Anonymous in acknowledgements of published article: Yes No

Recommendation to the editor

| | |
|---|---|
| Does the paper contain new data or new ideas or both of them? | **Yes** No |
| Are these up to international standards? | **Yes** No |
| Is the presentation clear? | **Yes** No |
| Does the author reach substantial conclusions? | **Yes** No |
| Is the length of the paper adequate? | **Yes** No |
| Is the language fluent and precise? | Yes **No** |
| Are the title and the abstract pertinent and understandable? | **Yes** No |
| Is the size of each figure adequate to the quantity of data it contains? | **Yes** No |
| Does the author give proper credit to related work and does he/she indicate clearly his/her own contribution? | **Yes** No |
| Would you cite this paper as a scientific contribution? | Very important **Fairly important** May have potential after additional work and resubmission    No potential value |

*Thank you for your constructive and helpful feedback. We really appreciate the all comments. Appropriate changes were made in the revised/final manuscript according all the suggestions.*

**Response to Referee 2.**

*Thank you very much for your consideration. We really appreciate the comments. Appropriate changes were made in the revised/final manuscript according your suggestions (specific comments are bellow).*

Referee #2

**Suggestions for revision or reasons for rejection (will be published if the paper is accepted for final publication)**

Even though I find this new version of the manuscript much improved compared with the first version, I have to reiterate my comments from my first revision. I start by saying that, I indeed think that the manuscript addresses a timely and interesting topic that certainly deserves attention from the scientific community. Mainly, I like the "multi-model" comparison of Polar Amplification between the tw hemispheres. In this direction, I congratulate the authors.

However, I feel like the authors didn't address most of my main comments, as follows:

1. In my opinion, the authors are using outdated data sets. We are already in the CMIP6 phase. This means that most of the models have already involved, corrected errors, improved parameterizations, etc, and so generated new data. The CMIP6 data sets are already available, and the computations performed by the authors are not that complex. They could easily be applied to the CMIP6 datasets. I am not saying that the CMIP5 data is not useful, but it would be more useful for the scientific community to see this multi-model comparison with the CMIP6 data. Or, it would be interesting to evaluate what are the differences between the Polar Amplification represented by the models from CMIP5 and CMIP6 phases. To be sure that I am not being impartial, I raised this discussion with other researchers involved in CMIP6 and this is indeed a kind of common sense in the community.

*Indeed, we have update the number of CMIP5 models, which now includes 15 models, as well as, we have include 17 CMIP6 models, for both, piControl and Abrupt 4xCO2 protocols.*

2. It doesn't help the fact that the authors are using a very limited number of models for a CMIP-like comparison. Since the authors are using relatively old data sets, the minimum that we could expect to see is a broader comparison with all models running the piControl and 4xCO2 experiments. To this point, the authors argued that there is even a Nature Climate Change paper (Harrison et al., 2015) that used only a few CMIP5 models. I don't see this as a valid argument for the following reasons: (i) this paper was published 5 years ago, so that I am not sure how many models had already contributed to CMIP5 by the date when the authors submitted their manuscript; (ii) back then, the CMIP5 was the current phase of CMIP, while now we are in the sixth phase; (iii) one of the objectives of Harrison et al. (2015) was, as I have mentioned above, to evaluate the "improvements in model performance between CMIP3 and CMIP5 in the simulation of large climate changes"

(see their goal n. 4); (v) as far as I understood, Harrison et al. (2015) used data from PMIP, a CMIP-endorsed project so that not all contributors to CMIP5 had run the paleo-simulations. This is not the case for the piControl and 4xCO2 experiments since many of the CMIP5 contributors have provided with those runs.

We agree. As pointed out in the response to the first point, we have now a sum of 32 models, which we believe encompasses the Polar Amplification phenomenon that we are focusing on in this manuscript.

3. I recall that in the first version of the manuscript the authors had based their conclusions on only three figures and, at this stage, it wasn't clear whether the manuscript was a short communication or a full article. So, I have suggested that the authors could also use the 1pctCO2 runs. In my opinion, this could bring robustness and make their study more complete. I am not sure what the authors think about this since they didn't present any answer to this comment, even though I have raised this suggestion both in the main and specific comments. In any case, I am not arguing that these analyses are a "must" for their study, but they would certainly make their manuscript looks like a full-article version (what it is still not clear for me).

Again, we agree with the reviewer's concerns and have restructured the manuscript accordingly. The new manuscript version, which includes more CMIP5 and CMIP6 models, presents a new range of Polar Amplification for both, Arctic and Antarctic. Also, differences in the seasonal values of Polar Amplification.

Based on the fact that I didn't see my first comments properly addressed, I am afraid that I can't give further recommendations on this manuscript's version. I am sorry that I can't be more positive at this stage.

---

## Author Response (AR3)

**Report #1**

Submitted on 10 Aug 2020
Anonymous Referee #2

**Anonymous during peer-review: Yes** No
**Anonymous in acknowledgements of published article: Yes** No

**Recommendation to the editor**

| | |
|---|---|
| Does the paper contain new data or new ideas or both of them? | **Yes** No |
| Are these up to international standards? | **Yes** No |
| Is the presentation clear? | **Yes** No |
| Does the author reach substantial conclusions? | **Yes** No |
| Is the length of the paper adequate? | **Yes** No |
| Is the language fluent and precise? | **Yes** No |
| Are the title and the abstract pertinent and understandable? | **Yes** No |
| Is the size of each figure adequate to the quantity of data it contains? | Yes **No** |
| Does the author give proper credit to related work and does he/she indicate clearly his/her own contribution? | **Yes** No |
| Would you cite this paper as a scientific contribution? | Very important **Fairly important** May have potential after additional work and resubmission    No potential value |

For final publication, the manuscript should be
**accepted as is.**
accepted subject to **technical corrections**.
accepted subject to **minor revisions**.
reconsidered after **major revisions**:
    I am willing to review the revised paper.
    I am **not** willing to review the revised paper.
**rejected**.

**Suggestions for revision or reasons for rejection** (will be published if the paper is accepted for final publication)
Please see "Recommendation to the editor"

Thank you very much for your consideration. We really appreciate and agree. Appropriate changes were made in each figure for  this revised/final manuscript according all the suggestions.